# Atlas of breast cancer infiltrated B-lymphocytes revealed by paired single-cell RNA-sequencing and antigen receptor profiling

Qingtao Hu [1,5], Yu Hong[1,2,5], Pan Qi[3,5], Guangqing Lu [1], Xueying Mai[1], Sheng Xu[3], Xiaoying He[3], Yu Guo[3], Linlin Gao[1], Zhiyi Jing[1], Jiawen Wang[1], Tao Cai [1] & Yu Zhang [1,2,4✉]

To gain mechanistic insights into the functions and developmental dynamics of tumor-infiltrated immune cells, especially B-lymphocytes, here we combine single-cell RNA-sequencing and antigen receptor lineage analysis to characterize a large number of triple-negative breast cancer infiltrated immune cells and report a comprehensive atlas of tumor-infiltrated B-lymphocytes. The single-cell transcriptional profiles reveal significant heterogeneity in tumor-infiltrated B-cell subgroups. The single-cell antigen receptor analyses demonstrate that compared with those in peripheral blood, tumor-infiltrated B-cells have more mature and memory B-cell characteristics, higher clonality, more class switching recombination and somatic hypermutations. Combined analyses suggest local differentiation of infiltrated memory B-cells within breast tumors. The B-cell signatures based on the single-cell RNA-sequencing results are significantly associated with improved survival in breast tumor patients. Functional analyses of tumor-infiltrated B-cell populations suggest that mechanistically, B-cell subgroups may contribute to immunosurveillance through various pathways. Further dissection of tumor-infiltrated B-cell populations will provide valuable clues for tumor immunotherapy.

[1] National Institute of Biological Sciences, Beijing, China. [2] Peking University-Tsinghua University-National Institute of Biological Sciences Joint Graduate Program, School of Life Sciences, Peking University, Beijing, China. [3] Xinxiang Central Hospital, Xinxiang, Henan, China. [4] Tsinghua Institute of Multidisciplinary Biomedical Research, Tsinghua University, Beijing, China. [5] These authors contributed equally: Qingtao Hu, Yu Hong, Pan Qi. ✉email: zhangyu@nibs.ac.cn

Although the roles of the immune system in tumor development and therapy were proposed long ago, they have only been recognized and mechanistically investigated recently[1,2]. Both the innate immune system (macrophages, neutrophils, mast cells, myeloid cells, dendritic cells, and natural killer (NK) cells) and adaptive immune system (T and B lymphocytes) contribute to the establishment of an immunosuppressive tumor microenvironment, which is one of the hallmarks of cancer[3]. More specifically, the adaptive immune system creates immunological memory after an initial response to a specific antigen (e.g., tumor antigen) and leads to an enhanced response during subsequent encounters with that antigen to provide long-lasting protection. Therefore, it plays essential roles during tumor development in the cancer immunosurveillance hypothesis and immunoediting hypothesis[4].

While it is well acknowledged that T lymphocyte-mediated adaptive cellular immunity has critical functions in the immune response for tumors[5], the roles of B lymphocytes in tumor development and therapy, both positive and negative, have only been proposed very recently and are still mostly controversial[6–13]. B cells may participate in tumor immunology through antibody production, antigen presentation, cytokine and chemokine production, and other immunoregulatory mechanisms[14–17]. For example, regulatory B (Breg) cells may play important roles in maintaining immune homeostasis by secreting cytokines (e.g., IL-10) and/or interacting with target cells[6,7,18,19]. Interestingly, three recent studies[20–22] demonstrated that B cells and tertiary lymphoid structures (TLSs) could be associated with better outcomes when individuals undergo immunotherapy, although the detailed cellular and molecular mechanisms still need to be defined.

Tumor-infiltrating T (TIL-T) cells have recently been characterized in several human cancer types by single-cell RNA sequencing (RNA-seq)[23–30]. However, a comprehensive atlas of tumor-infiltrating B (TIL-B) cells is still missing. B cells recognize antigens through B cell receptors (BCRs). The diversity of BCRs is generated by V(D)J recombination, somatic hypermutation (SHM), and class switch recombination (CSR) during B cell development and differentiation[31,32]. Antigen receptor repertoire analysis provides direct developmental lineage information[32,33]. To gain mechanistic insights into the functions and developmental dynamics of infiltrating immune cell, especially B cell, subgroups in breast cancer, we combined antigen receptor clonal lineage analysis and single-cell RNA-seq analysis to present an atlas of the BCR repertoire, clonal lineage, and transcriptional characteristics of TIL-B cells, which will serve as a foundation for studies of B cell tumor immunology.

## Results

Human breast cancer is a heterogeneous disease and contains several histologically different subtypes[34,35]. We analyzed infiltrated hCD45$^+$ immune cells in freshly isolated breast cancer samples by flow cytometry and found that the presence of infiltrated hCD19$^+$/20$^+$ B cells was significantly higher in triple-negative breast cancer (TNBC) than in other breast cancer subtypes (Supplementary Fig. 1). To characterize infiltrated immune cells, especially B cells in human breast cancer, we purified hCD45$^+$ cells from surgically isolated breast cancer tissues and corresponding peripheral blood mononuclear cell (PBMC) samples from six TNBC (TNBC1–6 and PBMC1–6), three luminal A breast cancer (LABC7–9 and PBMC7–9), and one HER2-positive breast cancer (HER2BC10 and PBMC10) treatment-naive patients (Fig. 1a). Among the samples from these ten patients, nine pairs (TNBC2–6, LABC7–9, HER2BC10, and PBMC2–10) were used to prepare 5′ single-cell RNA-seq libraries by droplet-based (10× Genomics) technology[26,36], while one pair (TNBC1

and PBMC1) was used for 3′ library construction (Supplementary Tables 1 and 2). In addition, single-cell immune repertoire information, including both BCRs and T cell receptors (TCRs), was also obtained for all 5′ libraries (Supplementary Table 3 and Supplementary Data 1).

Histological analysis, immunohistochemistry (IHC), and fluorescence in situ hybridization (FISH) were used to confirm the subtypes of breast cancer samples (Fig. 1a). Tumor-infiltrating lymphocytes (TILs) in the tumor stroma were assessed as percentages of occupied stromal areas according to the guidelines of the International TIL Working Group[37,38] (Fig. 1a).

After quality control and removing potential cell doublets[39], we obtained whole-genome RNA-seq data for 44,497 single cells with 1573 median genes per cell from ten tumor samples and 68,441 single cells with 1651 median genes per cell from the corresponding peripheral blood samples (Supplementary Table 2). Unsupervised clustering[40] at low resolution for these 112,938 single cells from all sequenced samples revealed four major cellular clusters, including T cells (marked by CD3D expression), B cells (CD20), NK cells (NKG7), and macrophages/monocytes/neutrophils (CD14) (Fig. 1b and Supplementary Fig. 2a). Cells from different samples contributed similarly to each cluster, suggesting a lack of sample batch effect (Supplementary Fig. 2b–d and Supplementary Table 4). The distributions of different infiltrated lymphocyte clusters in each tumor were heterogeneous across patients (Fig. 1c). For example, the percentages of B cells in tumor samples varied between 4.6% and 50.5%, with an average of 22.1%. T and NK cells constituted 21.4–73.7% (average 46.2%) and 0.5–5% (average 2.6%), respectively.

With the 10× Genomics 5′ V(D)J and gene expression chromium platform, we identified the rearrangement status of antigen receptor loci in each B and T cells. After quality control, we assembled rearranged BCRs (IGH) and TCRs (TCR α-β pair) for 26,401 single B cells and 44,621 single T cells, respectively, in all samples except TNBC1/PBMC1 (Supplementary Fig. 3 and Supplementary Table 3). Among them, there were 5951 and 16,485 single B cells containing a single productive IGH allele for BC and PBMC samples, respectively (Supplementary Fig. 3). As TNBC tumors have significantly more infiltrated B cells (Supplementary Fig. 1b), in the rest of the analysis, we focused mainly on B cells from the five TNBC patients (TNBC2–6 and PBMC2–6). Analysis of the single productive IGH rearrangements from 3695 infiltrated B cells in TNBCs and 8037 B cells from their corresponding PBMCs did not reveal significantly different VH, DH, or JH gene usage between the tumor and PBMC samples (Supplementary Fig. 4a). However, B cells infiltrated in TNBC contained a higher percentage of IGG-positive cells and a lower percentage of IGM- and IGD-positive cells than those in peripheral blood (Fig. 2a and Supplementary Fig. 4b). In agreement with this finding, the percentages of B cells with germline productive IGH alleles were significantly lower in TNBC tumors (Fig. 2a), suggesting that most of the B cells infiltrated in TNBC had encountered antigens and experienced BCR activation. SHM analysis[41,42] demonstrated that overall, TNBC-infiltrated B cells had significantly more IGH mutations than PBMCs (Fig. 2b). Interestingly, this difference was not only due to lower percentages of germline IGH-containing cells in tumor-infiltrated B cells. The SHM rates in non-germline and non-switched (IGM- and IGD-positive) B cells were also significantly higher in tumors than in PBMCs (Fig. 2b). Overall, there was no significant difference between TNBC and PBMC samples for IGH mutation types except G > T (Supplementary Fig. 4c–d).

We could identify individual B cells containing the same rearranged and mutated IGH alleles and B cell clones within which all B cells shared the same germline IGH (Supplementary

**a**

| Patient ID | Age (years) | Sex | SBR Grade | ER (IHC) | PR (IHC) | HER-2 (IHC) | HER-2 (FISH) | Ki-67 (IHC) | P53 (IHC) | Histology of Primary Tumor | Axillary Node Status | Stromal TIL value |
|---|---|---|---|---|---|---|---|---|---|---|---|---|
| TNBC1 | 54 | Female | III | - | - | + | NA | 70% | - | ductal | - | 40% |
| TNBC2 | 51 | Female | II~III | - | - | - | NA | 30% | - | others | - | 20% |
| TNBC3 | 47 | Female | II | - | - | + | NA | 50% | - | ductal | - | 30% |
| TNBC4 | 78 | Female | II | - | - | ++ | - | 30% | - | ductal | 1/10 | 10% |
| TNBC5 | 53 | Female | II | - | - | - | NA | 70% | + | ductal | - | 5% |
| TNBC6 | 63 | Female | II | - | - | + | NA | 40% | + | ductal | 1/3 | 60% |
| LABC7 | 56 | Female | II | 90% | 40% | ++ | - | 20% | - | ductal | - | 20% |
| LABC8 | 53 | Female | I | 90% | 50% | - | NA | 10% | - | ductal | - | 5% |
| LABC9 | 57 | Female | II | 80% | 80% | + | NA | 10% | - | ductal | 4/14 | 5% |
| HER2BC10 | 51 | Female | II~III | - | - | +++ | NA | 50% | + | ductal | - | 10% |

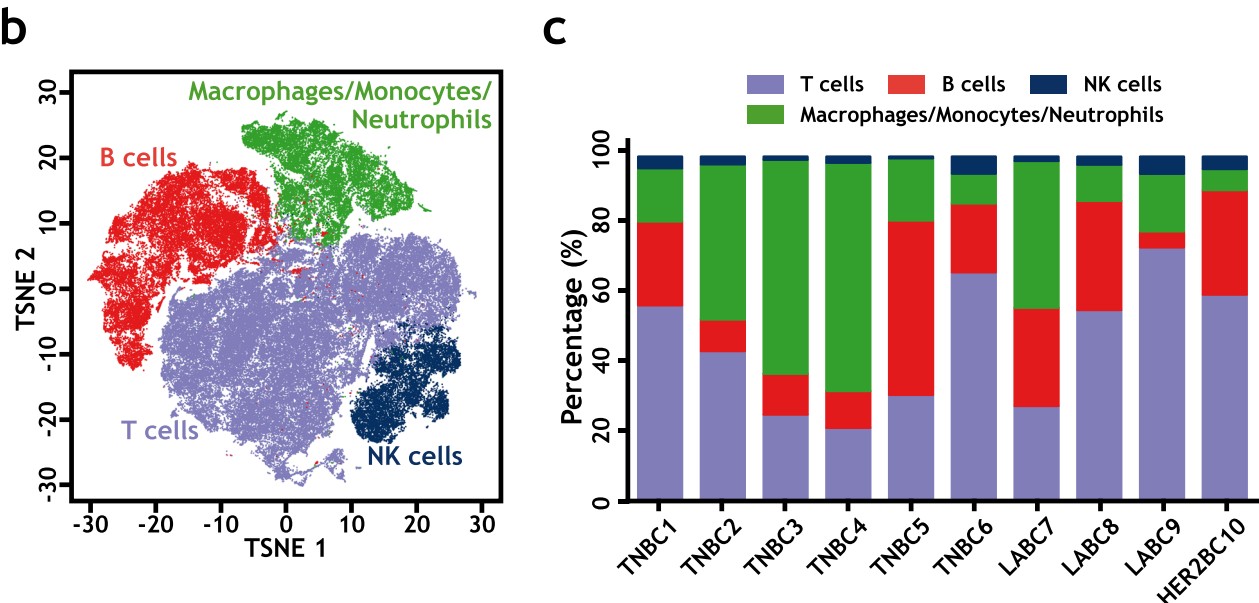

**Fig. 1 Heterogeneity of immune cells in human breast cancer patients. a** Clinical information of breast cancer patients involved in the current study. **b** The t-distributed stochastic neighbor embedding (t-SNE) projection of the immune cell atlas constructed from all patient samples. Each dot represents a cell, and major cell types are marked according to Supplementary Fig. 2a. **c** Distribution of major infiltrated immune cell types in each breast cancer sample. Source data are provided as a source data file.

Fig. 4e)[33]. As shown in Fig. 2c, TNBC-infiltrated B cells contained significantly higher percentages of B cell clones (19% vs. 7%), which were also larger and more complex than the clones in PBMCs (Fig. 2c and Supplementary Fig. 4f). The SHM rates in clones from TNBC samples were higher than those in clones from PBMC samples (Fig. 2d). Different tumor samples did not share common B cell clones.

Public clones were clonotypes shared by different cohorts[43,44]. When matching the clonotypes of TNBC patients to the dataset from Briney et al.[45], public clones were detected in both PBMC and TNBC samples, with 2.50% (193) in PBMCs and 2.23% (72) in TNBC, and the difference was not significant between TNBC and PBMCs ($p = 0.45$) (Supplementary Fig. 4g).

As in TNBC tumors, the infiltrated B cells in LABC and HER2BC tumors also showed similar BCR (*IGH*) phenotypes (Supplementary Fig. 5).

To further dissect the cellular diversity of infiltrated B cells in TNBC, we analyzed 2526 and 6106 single B cells that have both RNA-seq data and a single assembled productive *IGH* allele in tumor and peripheral blood samples, respectively. Unsupervised clustering of the combined samples (TNBC2–6 and PBMC2–6) at low resolution revealed four clusters with distinct transcriptional signatures (Fig. 3a, b, Supplementary Figs. 6–8, Supplementary Table 5 and Supplementary Data 2). Cells from different tumor and blood samples contributed to each cluster, suggesting a lack of sample batch effects and conserved differentiation processes (Supplementary Fig. 7c, d and Supplementary Table 5). These four clusters included naive B cells (*IGM*+ and *IGD*+), memory B cells (*CD27*+), plasma cells (*CD38*+), and *CD14*+ atypical B cells (*CD14*+). In agreement with the BCR results above, B cells infiltrated in TNBC samples were mostly memory B cells, whereas PBMC samples contained more naive B cells (Fig. 3c). The *IGH* SHM rates of memory B cells and

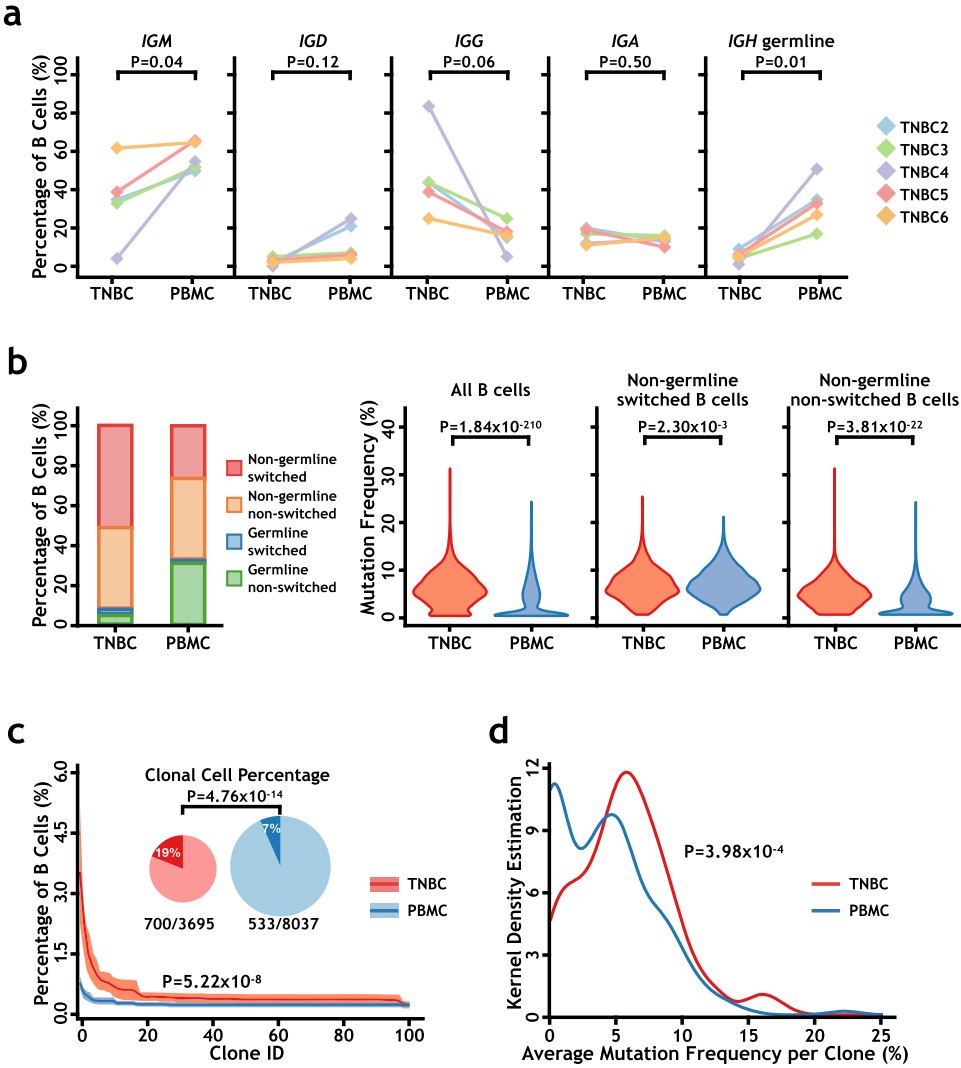

**Fig. 2 Single-cell *IGH* analysis of B cells in tumor and peripheral blood samples from TNBC patients. a** Distributions of *IGH* isotypes and germline *IGH* in B cells from triple-negative breast cancer (TNBC) and peripheral blood mononuclear cell (PBMC) samples. The *p* values were calculated by two-tailed paired Student's *t* test. **b** Comparisons of *IGH* class switching recombination (CSR) and somatic hypermutation (SHM) between B cells from TNBC and PBMC samples. The left bar plot presents the percentages of cells that have germline or non-germline and class-switched or non-switched *IGH* alleles. The violin plots from left to right show the comparisons of SHM rates for all B cells, for non-germline and class-switched B cells, and for non-germline and non-switched B cells. The *p* values were calculated by two-tailed Student's *t* test. **c** TNBC tumors contained significantly more and larger B cell clones. The percentages of clonal B cells in all B cells are presented by the upper pie charts, and the *p* value was calculated by two-sided Fisher's exact test using SPSS software. The size distribution of B cell clones shows that TNBC samples have larger B cell clones than PBMC samples. The *p* value was calculated by two-tailed Student's *t* test. **d** TNBC clones have higher *IGH* mutation frequencies than PBMC clones. The average *IGH* mutation frequencies of each clone are plotted versus the percent of clones with that mutation frequency. The *p* value was calculated by the two-sided Kolmogorov–Smirnov test. Source data are provided as a source data file.

plasma B cells were significantly higher than those of naive B cells (Supplementary Fig. 9a, b). Overall, the SHM rates of different B cell groups in TNBC were higher than those in PBMCs, except that plasma cells had higher SHM rates in PBMCs than in TNBCs (Supplementary Fig. 9c, d).

With the precise *IGH* V(D)J and SHM information for each single B cell, we were able to trace the proliferation and differentiation of B cells in tumors. After normalization, in a heat map to demonstrate the intra- and intercluster distribution of B cell clones with the same *IGH* germline (Fig. 3d), we found that in TNBC infiltrated B cells, memory B cells had the most clones which share the same inferred *IGH* germline, and plasma cells had the most clones with the same detected *IGH* sequence.

At higher resolution, unsupervised clustering further separated 8632 single B cells into thirteen clusters with distinct

transcriptional signatures (Fig. 4a, b, Supplementary Fig. 8b, Supplementary Fig. 10a–d, Supplementary Table 6, and Supplementary Data 3). By combinatorically analyzing the expression of known B cell marker genes[46–48] and productive *IGH* sequences, we annotated these 13 subgroups. They included naive B cells (C1 and C2), *IGM+CD27+* memory B cells (C3, C4, C6, and C7), *IGM+CD27−* atypical memory B cells (C5), class-switched memory B cells (C8–C10), plasma cells (C11), germinal center B cells (C12), and *CD14+* atypical B cells (C13). The composition of those B cell clusters was different between tumor and PBMC samples (Fig. 4c and Supplementary Fig. 11).

The presence of germinal center B cells (C12), *CD1C*<sup>High</sup> memory B cells (marginal zone B cells, C6)[49], and various class-switched memory B cell subgroups (C8–C10) in TNBC tumors suggested locally ongoing CSR and SHM in potential TLSs[50,51]

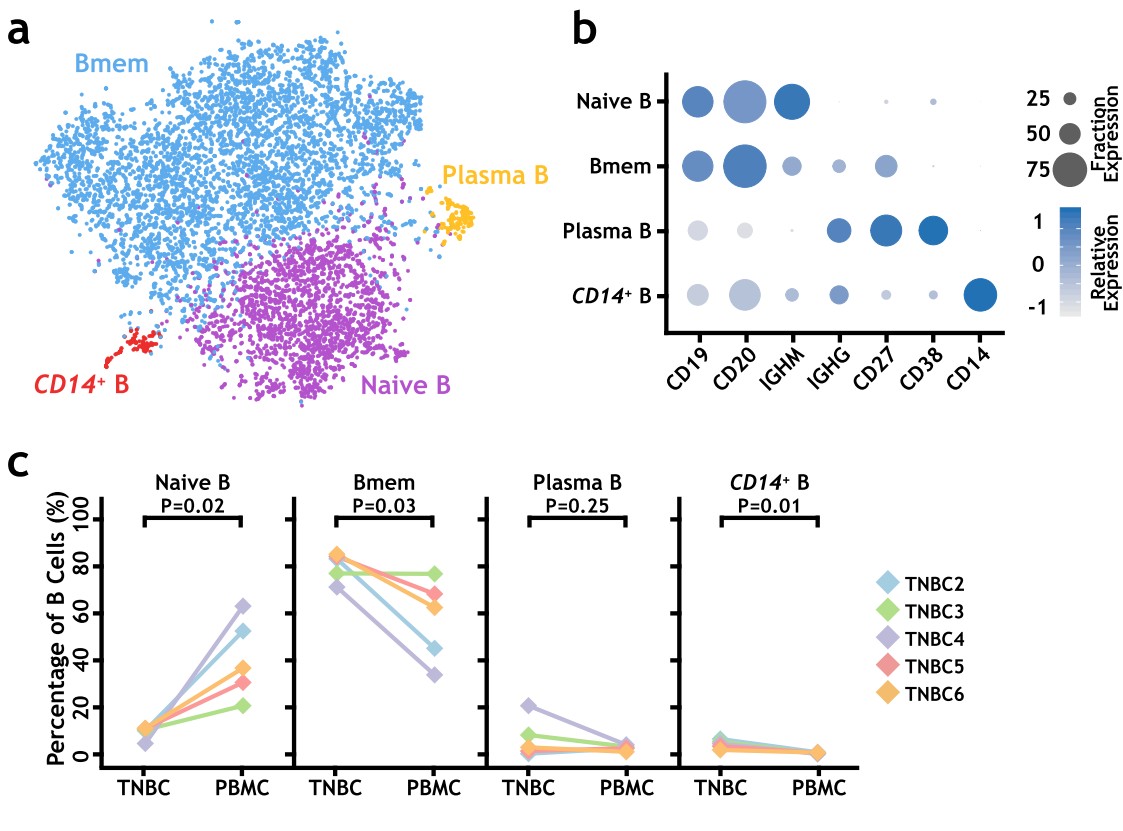

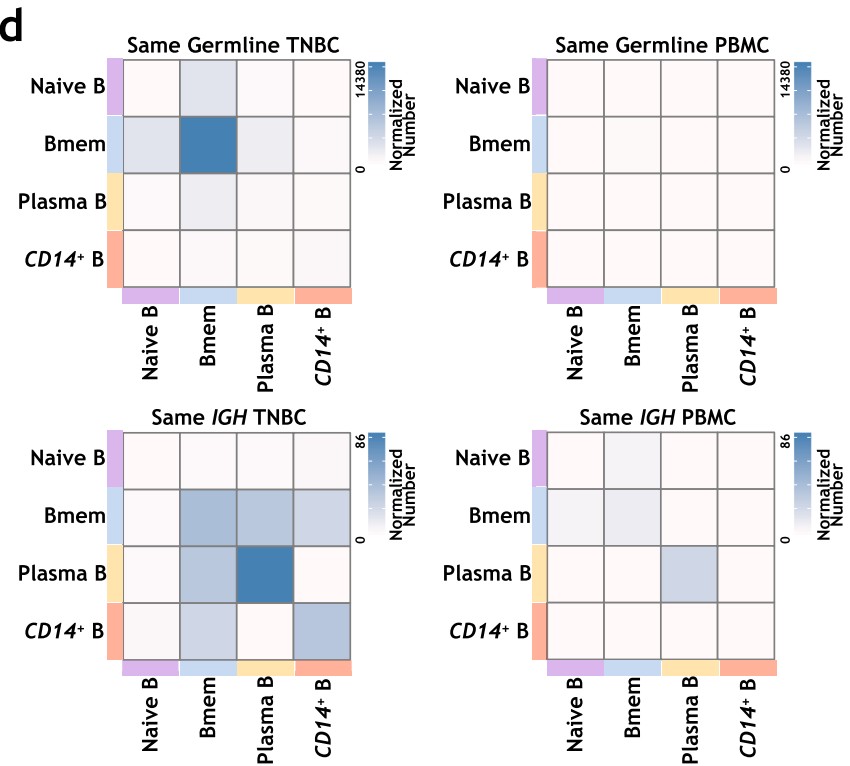

**Fig. 3 Single-cell transcriptome analysis and *IGH* lineage analysis of B cells from TNBC patients at low resolution. a** The t-SNE projection of 8632 B cells from TNBC patients shows four major cellular clusters. **b** Selected marker genes to define each B cell cluster. **c** The distributions of four clusters in each patient sample. The *p* values were calculated by two-tailed paired Student's *t* test. **d** Heat maps show the distribution of the same *IGH* germline events (upper panel) and the same *IGH* sequence events (lower panel) among the four B cell clusters for TNBC and PBMC samples. The same germline events were calculated using clonally related B cells, and the same *IGH* sequence events were calculated using B cells with the same observed *IGH* sequences. Then, the event numbers were normalized by the cell numbers of related B cell clusters and plotted in the heatmap. See "Paired BCR and single-cell RNA-seq data analyses" in the "Methods" section for details. Source data are provided as a source data file.

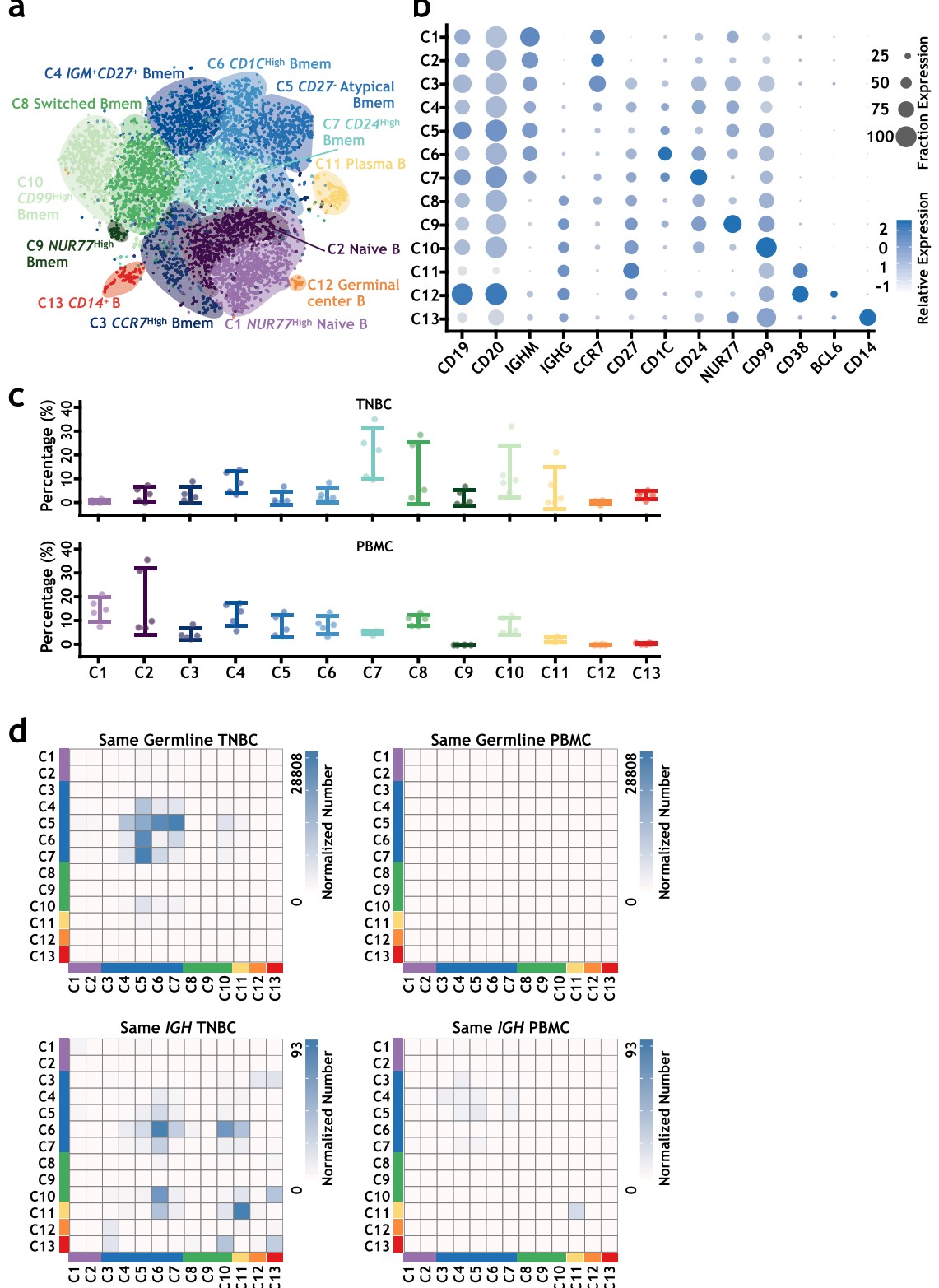

**Fig. 4 Single-cell transcriptome analysis and *IGH* lineage analysis of B cells from TNBC patients at a higher resolution. a** The t-SNE projection of 8632 B cells from TNBC patients shows 13 major cellular clusters at a higher resolution. **b** Selected marker genes to define each B cell cluster. **c** The distributions of 13 clusters in each patient sample (upper panel: PBMC; lower panel: TNBC). X represents B cell clusters, and Y represents the cell percentage in each sample. Each dot represents one sample, and error bars represent the mean ± standard deviation (SD) of the five samples. **d** Heat maps show the distribution of the same germline event number (upper panel) and the same *IGH* sequence event number (lower panel) among the 13 B cell clusters for TNBC and PBMC samples. The same germline events were calculated using clonally related B cells, and the same V(D)J sequence events were calculated using B cells with the same observed *IGH* sequences. Then, the event numbers were normalized by the cell numbers of related B cell clusters and plotted in the heatmap. See "Paired BCR and single-cell RNA-seq data analyses" in the Methods section for details. Source data are provided as a source data file.

within tumors. We identified T follicular helper (Tfh) cells in the T cell clusters (Supplementary Results). Moreover, germinal center (C12) B cells expressed high levels of activation-induced cytidine deaminase (*AICDA*) and *MKI67*, and demonstrated a strong proliferation gene signature (Supplementary Fig. 10e). We also confirmed TLS structures in TNBC samples by immunohistochemical staining for CD20, CD3, CD21, Ki-67, and PNAd (Supplementary Fig. 12a) and multiplex immunofluorescence staining for CD20, CD3, and CD21 (Supplementary Fig. 12b). By multiplex immunofluorescence staining for CD20, CD3, and Ki-67, we also identified the putative proliferating germinal center B cells in TNBC samples (Supplementary Fig. 12c). All these results suggested the existence of functionally active germinal centers in TNBC tumor tissues[52].

We also tried to identify the potential Breg cells in TNBC. Although *IL-10* expression could be well detected in monocytes and macrophages, we could not identify a specific *IL-10*-expressing B cell population (Supplementary Fig. 13a, b). This was similar for granzyme B (*GZMB*) and *PDCD1* expression, two other Breg marker genes (Supplementary Fig. 13c–f).

As in TNBC, the TIL-B subgroup distributions in LABC and HER2BC were similar. The memory B cell subgroup was the dominant B cell subgroup in cancer, and the percentage of memory B cells was significantly higher than that in PBMCs, while the percentage of naive B cells was significantly lower than that in PBMCs (Supplementary Fig. 14).

To test whether the TIL-B subgroup distribution is cancer type-specific, we mapped TIL-B cell data from colon cancer ($n = 2$), lung cancer ($n = 2$), and renal cancer ($n = 3$)[53] to the 4 subgroups or 13 subgroups of B cells. The memory B cell subgroup was the main TIL-B subgroup in all of the above cancer types, and the percentage distribution of the 4 B cell subgroups was similar among different cancer types (Supplementary Fig. 15a). For the percentage distribution of the 13 B cell subgroups, samples from the same cancer type tended to be clustered together with each other in the subject clustering analysis, though more samples for each cancer type were needed to further confirm this clustering result (Supplementary Fig. 15b, c).

In TNBC samples, mainly non-switched memory B clusters (C4–C7) were significantly involved in *IGH* BCR clonal trees (Fig. 4d). In particular, the $CD27^-$ atypical memory B cells (C5) had the highest intracluster diversity, suggesting more SHMs within this cluster. In TNBC, $CD27^-$ atypical memory B cells also shared the same germline significantly with $IGM^+CD27^+$ memory B cells (C4), $CD1C^{High}$ memory B cells (C6), and $CD24^{High}$ memory B cells (C7).

In the heat map for the same *IGH* sequences within and between different B cell clusters (Fig. 4d), plasma cells (C11) had the highest percentages of cells with the same *IGH* allele within clusters in both TNBC and PBMC samples, suggesting their clonal expansion. However, these plasma cells (C11) had the lowest proliferation score (Supplementary Fig. 10e). Interestingly, in TNBC, $CD1C^{High}$ memory B cells (marginal zone B cells, C6) also showed significantly higher clonal expansion than the other clusters. In addition, $CD1C^{High}$ memory B cells shared significant amounts of the same *IGH* with other B cell clusters, such as $CD99^{High}$ memory B cells (C10), plasma cells (C11), and non-switched memory B clusters (C4–C7).

Recently, a subpopulation of T cells, which is associated with active proliferation and tumor reactivity, has been revealed by single-cell analysis of infiltrated T cells in human melanoma[25]. In our TNBC samples, some infiltrated T cells also showed significant ongoing proliferation, as evaluated by both cell proliferation and cell cycle signatures (Supplementary Fig. 16a, c). In contrast, most TNBC-infiltrated B cells, except germinal center B cells (Supplementary Fig. 16), were not actively proliferating.

The roles of B lymphocyte infiltration in breast cancer and other types of cancers remain controversial[9,10]. In most studies, the presence of infiltrated B lymphocytes was determined by IHC of pan-B cell markers such as CD20. We hypothesized that distinct infiltrated B cell clusters might have diverse functions in tumor immunology and that the combinational transcriptional signature based on all the differentially expressed marker genes (Supplementary Table 7) of individual B cell clusters might be a better indication of their presence in tumors. Indeed, using the expression data and clinical information of TNBC patients from the METABRIC consortium, we found that the transcription signatures of naive B cells and memory B cells were significantly associated with improved overall survival and disease-free survival in TNBC patients (Fig. 5a). More importantly, compared with classic single B cell marker *CD20*, those B cell signatures showed much stronger hazard ratios (HRs) in both univariable and multivariable analyses for TNBC patients (Fig. 5b and Supplementary Fig. 17), suggesting that they provided better prognostication than *CD20* alone.

TNBC tumor samples had a higher expression of naive and memory B cell signatures than the other subtypes of breast cancers (Supplementary Fig. 18). These two B cell signatures were not associated with better overall survival and disease-free survival in non-TNBC patients (Supplementary Data 4). We further analyzed the expression levels of the naive and memory B cell signatures in various cancer types in The Cancer Genome Atlas (TCGA) consortium (https://www.cancer.gov/tcga). As shown in Supplementary Fig. 19, both B cell signatures showed variable expression levels in different kinds of cancer types. Interestingly, these B cell transcription signatures could be associated with prognosis in distinct tumor types. For example, the memory B cell signature demonstrated a significant association with better overall survival and stronger HRs in patients with cervical squamous cell carcinoma and endocervical adenocarcinoma (CESC), sarcoma (SARC), skin cutaneous melanoma (SKCM), and uterine corpus endometrial carcinoma (UCEC) (Supplementary Fig. 20 and Supplementary Data 5).

## Discussion

The immune microenvironment plays essential roles in tumor development and therapy[1,2,54]. While T cells have been extensively studied and therapeutically targeted in tumor immunology, the roles of B cells have only been noticed recently[6–10]. TIL-B cells have been identified and are associated with better or worse prognosis in various human tumors[10], such as ovarian cancer[55,56], breast cancer[57,58], lung cancer[59,60], colorectal cancer[61], and renal cell cancer[62]. In particular, the association of TIL-B cells with the prognosis of human breast cancers has been very controversial[9], which might be due to the heterogeneity of both breast cancer patients and B cell subtypes[13,63–65]. We focused on TNBC, the most immunogenic breast cancer subtype, and dissected the tumor-infiltrated B cell populations by paired single-cell antigen receptor repertoire and whole-transcriptome sequencing. We presented a comprehensive single-cell analysis of TNBC-infiltrated B cells and found that TNBC-infiltrated B cells showed more mature and memory B cell characteristics. These TNBC-infiltrated memory B cells had higher clonality and extensive *IGH* CSR and SHMs, which likely happened within tumors and experienced tumor antigen recognition. Our results also confirmed the existence of functionally active germinal centers in TNBC tumors. On the other hand, we could not detect obvious regulatory B cell populations characterized by *IL-10* expression in TNBC. The TIL-B gene transcription signatures based on single-cell RNA-seq results could be associated with the improved survival of TNBC patients and provided better prognostication than classic B cell marker (*CD20*).

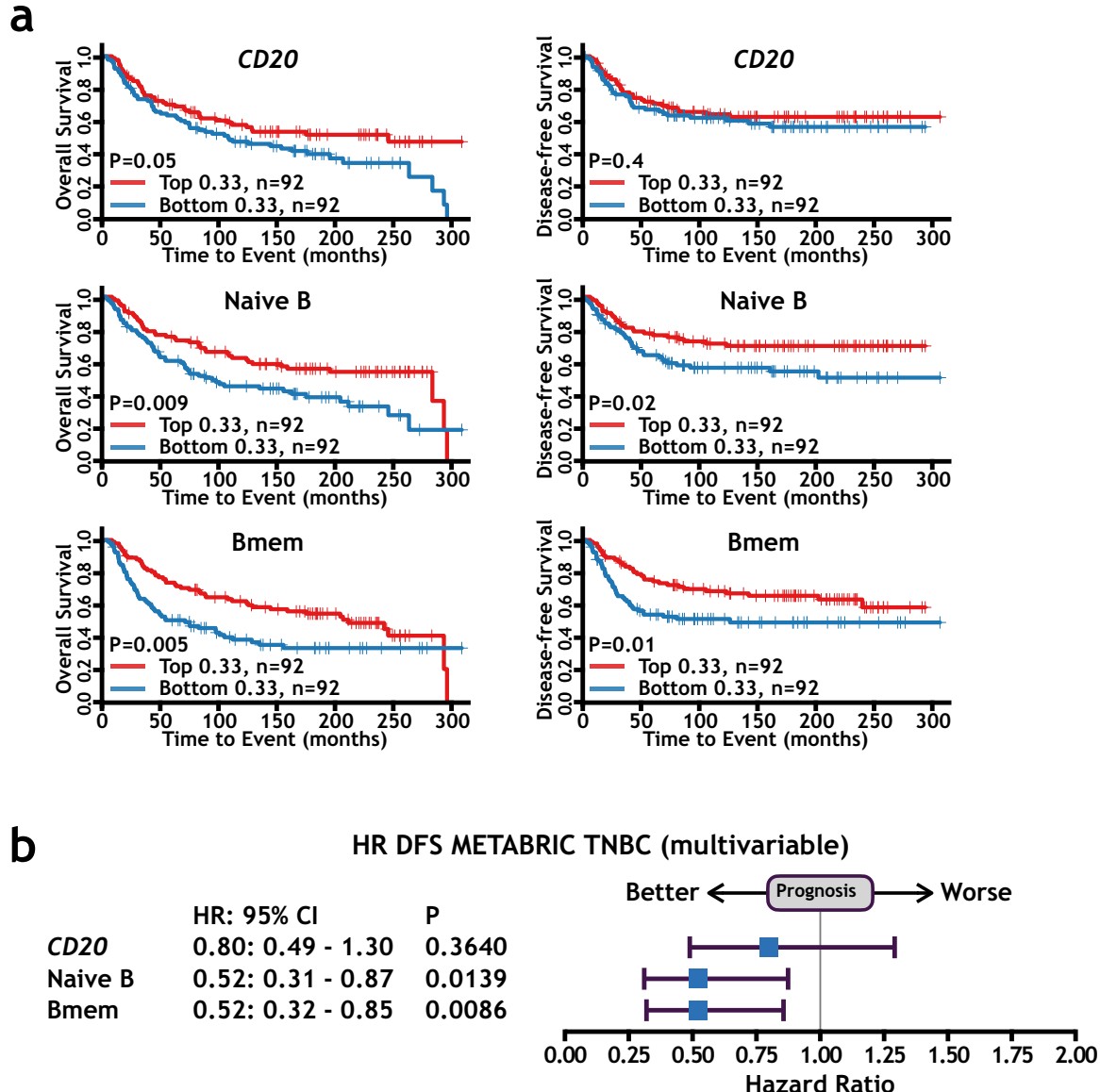

**Fig. 5 The B cell signatures based on the single-cell RNA-seq results are significantly associated with improved survival in TNBC patients and provide better prognostication than classic single B cell markers. a** Kaplan–Meier survival curves for the overall survival (left panels) and disease-free survival (right panels) of METABRIC TNBC patients according to single gene expression (*CD20*) or combined gene transcription signatures (naive B cells and memory B cells). The 279 TNBC patients were sorted according to the expression levels of the gene or gene signatures, and the top 33% ($n = 92$) vs. bottom 33% ($n = 92$) of patients were used to generate survival curves ("Methods"). The *p* values were calculated by the two-sided log-rank test. **b** Prognostic effect of *CD20*, naive B signature, and memory B signature in METABRIC TNBC patients. Forest plot shows HRs (the center blue squares) and 95% confidence intervals (horizontal ranges) derived from Cox regression survival analyses of disease-free survival in multivariable analyses adjusted for lymph node status, tumor size, age at diagnosis, and histological grade. Data were presented as HR and 95% confidence intervals.

B cells could positively participate in tumor immunology through various immunoregulatory mechanisms[6–8,17]. Therapeutic tumor-reactive antibodies mediate tumor lysis by Fc-driven innate immune effector function. Moreover, recent evidence suggests that tumor antigen-specific antibodies could also drive therapeutic T cell responses[15]. A tumor antigen-specific antibody was necessary for maximal antitumor efficacy in combinational immunotherapy that engaged both innate and adaptive immune responses[16,66]. In animal models, the adoptive transfer of tumor-reactive B cells could lead to tumor regression by secreting antibodies and activating T cell immunity[14,67]. Activated B cells could also work efficiently as antigen-presenting cells (APCs) to stimulate the tumor antigen-specific T cell response[68,69]. On the other hand, whether depleting endogenous B cells could suppress or enhance tumor immunology is still controversial[17,70,71]. We found that TNBC plasma cells had significantly higher expression levels of immunoglobulin (*IG*) genes than PBMC plasma cells (Supplementary Fig. 21). Gene set enrichment analysis (GSEA) also revealed that both genes involved in antigen processing and presentation and interferon-alpha response genes showed significantly higher expression in TNBC B cells than in PBMC B cells (Supplementary Fig. 22). Interestingly, the relative expression levels of those two gene sets were significantly correlated among various B cell subgroups. Finally, in the METABRIC TNBC dataset, the expression levels of naive and memory B cell signatures could be highly correlated

with the expression of general T cell marker genes (such as *CD3G* and *CD8A*) and gene signatures of various T cell groups (such as Tfh and tissue-resident memory T (Trm) cells) (Supplementary Fig. 23 and Supplementary Table 8). Similar results were also observed in breast and ovarian cancer patients in TCGA[57], suggesting that TIL-B cells may cooperate with TIL-T cells in tumor immunology.

Breg cells were originally defined as *IL-10*-expressing B cells, which can regulate the functions of other immune cells, especially T cells[18,19,72]. Although Breg cells have been identified in various human cancers, including breast cancer[19,73–75], we could not clearly define Breg cell populations by *IL-10* expression. By analyzing macrophages and monocytes that also expressed *IL-10* (Supplementary Fig. 13), we ruled out the possibility that the technical limitations of single-cell sequencing might lead to the inefficient detection of *IL-10* expression in B cells. In addition, a few *IL-10*-expressing B cells could not form distinct clusters from the rest of the B cells. Moreover, we could not find a negative association between the gene expression signatures of any B cell subgroups and the prognosis of TNBC patients. These results suggest the absence of Breg cells in TNBC patients. Interestingly, Breg cells were also not detected in recent TNBC mouse models[17].

The identification of the origin and development dynamics of those TIL-B cells in TNBC will be the next important question. Here, we demonstrated that TIL-B cells in TNBC showed various significant differences from peripheral circulating B cells. In particular, infiltrated memory B cells share some characteristics with recently defined mouse lung resident memory B (BRM) cells[76]. For example, they had clonal BCRs that did not significantly overlap with those of circulating B cells in peripheral blood. They also had higher BCR clonality. BCR lineage analysis of the clonal trees consisting of both PBMC and TNBC B cells revealed that PBMC B cells dominantly occupied upstream of TNBC-infiltrated memory B cells (Supplementary Fig. 24). These results suggested that TNBC-infiltrated B memory cells might have developed and differentiated within tumors. However, to completely confirm that they are indeed BRM cells, in contrast to circulating memory B cells, further work is needed. For example, the identification of the potential tumor antigens for these dominant BCR clones[77] will be the next urgent step to understand the origin of those tumor-infiltrated B cells in TNBC.

Dissecting the heterogeneity of tumor-infiltrated B cell subgroups provided essential information to functionally reveal their roles in the development and treatment of cancers. Further studies are needed to explore the mechanisms leading to the clear association of tumor-infiltrated B cells with better survival in TNBC patients. Pathways that regulate the infiltration, expansion, and differentiation of TIL-B cells could be used as new targets for TNBC immunotherapy[78,79].

Although we thoroughly analyzed only TNBC-infiltrated B cells here, we also obtained single-cell transcriptional data for 88,254 other immune cell types and single-cell TCR results for 58,095 T cells (HRA000477). These data will be invaluable to understand the overall tumor immune environment and potential interactions between different immune cell types, as well as to identify new immunotherapy targets. Further studies on the characterization of all infiltrated immune cell types in TNBC and the mechanisms by which TNBC escapes immunosurveillance in human patients will provide essential guidance for the immunotherapy of TNBC. Our analysis and experimental strategies could also be widely applied to B cell analysis in other tumor types.

## Methods

**Reagents and antibodies**. The fluorescence-activated cell sorting (FACS) antibodies used in this study included PE anti-human CD45 (BioLegend, Cat# 304008) (1:100 dilution), APC anti-human CD3 (BioLegend, Cat# 300312) (1:100 dilution),

FITC anti-human CD20 (BioLegend, Cat# 302304) (1:100 dilution), and FITC anti-human CD19 (BioLegend, Cat# 302206) (1:100 dilution). The IHC and multiplex immunofluorescence antibodies used in this study included CD3 (Celnovte, CCM-0332) (ready to use, no dilution), CD20 (Celnovte, CCM-0461) (ready to use, no dilution), CD21 (Celnovte, CCR-0471) (ready to use, no dilution), PNAd (BD, 553863) (1:25 dilution), and Ki-67 (Celnovte, CKM-0032) (ready to use, no dilution).

**Patient samples**. This study was approved by the human research ethics committee of the Xinxiang Central Hospital (Xinxiang, P.R. China). All participating patients provided written informed consent. Breast cancer tissues were collected during surgery from patients who had not undergone any chemotherapy or other treatments. Peripheral blood samples were collected from the same patients.

PBMCs were isolated using Lymphoprep™ (Sigma-Aldrich) solution according to the manufacturer's instructions. Fresh breast tumor samples were cut into small pieces and gently triturated with a 5-ml syringe plunger on a 70 μm Cell-Strainer (BD) in RPMI-1640 medium (Invitrogen) with 2% fetal bovine serum (FBS) on ice until uniform cell suspensions were obtained. The cells were subsequently passed through cell strainers and centrifuged at 400×g for 10 min. The cell pellets were resuspended in 6 ml of RPMI-1640 medium supplemented with 2% FBS. The next steps were the same as those for peripheral blood sample preparation.

**Patient diagnosis**. Breast cancer patients were diagnosed by disease history and mammogram in Xinxiang Central Hospital, Henan, China. Then, tissues were collected from patients undergoing surgery and subjected to histological and immunohistochemical analysis for HER2, ER, PR, and Ki67. The FISH analysis was conducted to confirm the *HER2* status when the IHC result was two-plus (++). The breast cancer subtypes were defined by the IHC and FISH results as TNBC (HER2−, ER−, PR−), HER2BC (HER2+, ER−, PR−), LABC (HER2−, ER+, and/or PR+, Ki67low), and luminal B breast cancer (LBBC) (HER2+/−, ER+ and/or PR+, Ki67high).

The sample exclusion criteria were as follows: (1) patients treated with any chemotherapy before surgery were excluded; (2) samples with a small number of B cells by FACS analysis were excluded; and (3) samples that failed the quality control during single-cell RNA-seq library construction were excluded from further analysis. More information on the patients is shown in Supplementary Table 1.

**Histological analysis**. Histological and immunohistochemical analyses of the tumor samples were performed by central labs at Xinxiang Central Hospital. Images were acquired by Olympus VS120. TILs in the tumor stroma were assessed as percentages of occupied stromal areas according to guidelines of the International TIL Working Group[37].

**Immunofluorescence staining**. A tyramide system amplification (TSA) was used for the CD3/CD20/CD21 and CD3/CD20/Ki-67 three-plex staining. The incubation with TSA reagent was performed after the incubation of the horseradish peroxidase-conjugated polymer and was followed by antibody stripping at 97 °C for 10 min. This protocol was repeated for the second and third primary antibodies and corresponding polymer incubations. The dilutions used for the TSA are 1:300 for CY3-TSA, 1:500 for FITC-TSA, and 1:500 for CY5-TSA. For all the fluorescent stainings, the nuclei were stained with DAPI solution at 2 μg/ml for 10 min. The slides were scanned with a Pannoramic Scanner (Pannoramic P250; 3D HISTECH).

**FACS analysis and sorting**. Lymphocytes prepared by Lymphoprep™ (Sigma-Aldrich) solution were resuspended in FACS sorting buffer (phosphate-buffered saline supplemented with 2% FBS) and stained with the indicated FACS antibodies. The DAPI-negative and hCD45-positive cells within the singlet-gated subpopulation were sorted by BD FACSAria II. Data were acquired and analyzed by FlowJo (version10).

**Single-cell RNA-seq and TCR/BCR library construction**. The FACS-sorted viable hCD45+ cells were resuspended at 5 × 10^5–1 × 10^6 cells/ml with final viability of >85%. Single-cell library preparation was carried out using the 3′ library v2 or 5′ V(D)J and gene expression platform as per the 10× Genomics protocol (10× Genomics, Pleasanton, CA, USA). Cell suspensions were loaded onto a Chromium Single-Cell Chip along with the reverse transcription master mix and single-cell gel beads, aiming for 2000–10,000 cells per channel. Following the generation of single-cell gel bead-in-emulsions (GEMs), reverse transcription, and amplification were performed. Then, amplified cDNAs were purified and sheared. Sequencing libraries were generated with a unique sample index for each sample. Libraries were sequenced by Illumina HiSeq ×10 or NovaSeq.

**Quality control and filtering of single-cell RNA-seq data**. The 10× Cell Ranger package (version 2.1.1, 10× Genomics) was used to demultiplex cellular barcodes, map reads to the hg38 reference assembly (v1.2.0, 10× Genomics), and generate a feature-barcode matrix (the number of unique molecular identifiers (UMIs) associated with a feature (row) and a barcode (column)). The 20 single-cell RNA-

seq libraries were sequenced to average 74,161 (40,341–191,921) paired-end reads per cell with 88.3% (80.2–94.7%) sequencing saturation (Supplementary Table 2). To remove multiple captures, which is a major concern in microdroplet-based experiments, we excluded the top $X$ cells with the highest pANN score calculated by DoubletFinder (version 1.0.0)[39] for each library separately, where $X$ was inferred as formula (1): $X = (0.000879*N + 0.702)*0.01*N$ with linear fitting using the 10× platform data from Supplementary Fig. 1a [36], where $N$ represents the cell number detected by Cell Ranger. Cells with numbers of detected genes <200 or >5000 and cells with >50% of the UMI counts belonging to mitochondrial genes were also omitted. Finally, 112,938 cells from a total of 20 libraries were retained for downstream analysis, with 4811 median UMIs and 1600 median genes per cell.

**Canonical correlation analysis (CCA), dimensionality reduction, and clustering**. After quality control and filtering, the feature-barcode matrices of each library were processed by Seurat (version 2.3.4)[40] for normalization, dimension reduction, batch effect removal, graph-based clustering, cluster-specific marker gene detection, and visualization. Library-size normalization to each cell was performed by NormalizeData. The variability of the numbers of UMIs was regressed out by ScaleData. The variable genes were calculated by FindVariableGenes. Then, all 20 libraries were combined together using diagonal CCA by RunMultiCCA. The genes used for correlation component (CC) calculation were the combination of the top 2000 dispersed genes for each library (a total of 3175 genes appeared in at least two samples). The cells were then aligned with AlignSubspace using 10 CC dimensions. The CC number was determined by inspecting the results of DimHeatmap. Using CalcVarExpRatio to calculate the percentage of variance explained by the 10 CCs for each cell, there were 95.68% cells with 50% or more variance explained. FindClusters was used to cluster cells using the 10 aligned CCs at a resolution of 0.1 (total four clusters). A set of canonical markers (CD3D, CD20, CD14, and NKG7) was expressed at high levels in distinct clusters (Supplementary Fig. 2a). The clustering results were visualized with t-distributed stochastic neighbor embedding (t-SNE) dimensionality reduction using RunTSNE (10 aligned CCs) and ggplot2 R package.

**BCR and TCR data quality control and analysis**. The raw sequencing reads of the BCR and TCR libraries were processed by Cell Ranger VDJ pipeline (version 2.1.1, 10× Genomics) with the default parameters to assemble contigs that represent the best estimate of transcript sequences in each cell. The contigs of IGH, IGK, and IGL were assembled from the BCR libraries. Here, we only used sequences of IGH without considering IGK or IGL to include more B cells in further analyses[80]. Cells with IGH contigs were defined as BCR-positive cells.

The annotation results of IMGT/HighV-QUEST (V1.5.7.1) for those B cells were processed by Change-O (V0.3.12)[41] and custom scripts to select cells with a single productive IGH. Productive IGHs were selected using ParseDb.py with parameters -f FUNCTIONAL -u T. If there were more than one productive IGH detected within one cell, the most abundant IGH sequence was kept when the UMI counts of the most abundant IGH sequence were more than 10-fold of the second most abundant IGH sequence. Finally, there were 22,436 (86%) B cells with a single productive IGH and 3736 (14%) B cells with multiple productive IGHs. The B cells with multiple productive IGHs were removed before further analysis, as they had significantly more UMIs than single productive IGH B cells, suggesting that they were B cell doublets (Supplementary Fig. 3a, b). We obtained 11,732 cells with a single productive IGH and 8632 cells with both transcriptome and IGH sequence data for TNBC2–6 and PBMC2–6 samples (Supplementary Table 3 and Supplementary Data 1). The assembled contigs (filtered_contig.fasta files from Cell Ranger) were processed by IMGT/HighV-QUEST[42] to assign V(D)J germline segments to each contig using the default settings for the human IGH allele. Clonally related B cells were defined as those sharing the same germline sequences, which means B cells within the same clone have the same V, D, J usage and the same length of junctions for their IGH sequences. The SHM number and frequency were calculated using only V and J regions (the D region and junctions were not included). Lineage trees were inferred using PHYLIP version 3.69[81].

In the TCR libraries, cells with both TRA and TRB contigs assembled were defined as TCR-positive cells. Cells with BCR and TCR double positivity were detected (1195 cells, 4.5% of BCR-positive cells or 2.7% of TCR-positive cells). They were removed before B cell subgrouping analysis, as they had significantly more UMIs than BCR single-positive cells, TCR single-positive cells, and BCR/TCR double-negative cells, suggesting that they were doublets (Supplementary Fig. 3a, c). After filtration, there were 26,172 BCR-positive, TCR-negative B cells.

**Public B cell clone analysis**. Public B cell clones were defined as clonotypes shared by different cohorts. We downloaded the dataset from Briney et al. (https://github.com/briney/grp_paper), which includes more than one hundred million clonotypes from ten blood samples, and a B cell clonotype was defined as a collection of sequences using the same V and J genes and encoding an identical CDRH3 amino acid sequence[45]. The percentage of public clones was defined as the total number of unique clonotypes shared between the current study and the dataset from Briney et al. divided by the number of unique clonotypes in the current study. Chi-square tests were performed using the chisq.test function in R for the number of public clones and the number of nonpublic clones.

**B cell clustering analysis for TNBC2–6 patients**. After the quality control and filtering of single-cell RNA-seq data, B cells with a single productive IGH in BCR libraries and TCR negativity in TCR libraries were selected for B cell subgrouping analysis (8632 cells for TNBC2–TNBC6 patients). For those B cells in each library, library-size normalization to each cell was performed by NormalizeData. The variability of the numbers of UMIs was regressed out by ScaleData. The variable genes were calculated by FindVariableGenes. RunMultiCCA was used to combine the B cells from all libraries, and 10 CC dimensions were used for AlignSubspace. FindClusters was used to cluster cells using the 10 aligned CCs at a resolution of 0.1 for low-resolution grouping (Fig. 3) and at a resolution of 1.0 for high-resolution grouping (Fig. 4). The CC number was determined by inspecting the results of DimHeatmap. Using CalcVarExpRatio to calculate the percentage of variance expressed by the 10 CCs for each cell, there were 94.24% cells with 50% or more variance explained. The clustering results were visualized with t-SNE dimensionality reduction using RunTSNE (10 aligned CCs) and ggplot2 R package. Canonical markers (CD19, CD20, CD27, CD38, IGHD, IGHM, and CD14) were expressed at high levels in distinct clusters (Supplementary Fig. 6a). The marker genes of each cluster were detected by the FindAllMarkers function with the default parameters, except min.pct was set to 0.25. The genes were ranked by their fold-change (from largest to smallest), and the top 10 marker genes of each cluster are shown in the heatmap in Supplementary Fig. 8. T cell clustering analysis for TNBC2–6 patients was described in Supplementary Results, Supplementary Data 6, and Supplementary Table 9.

We defined the signature genes of each cell group for survival analysis[24]. Cells from TNBC tumor samples were used to identify signature genes defined by DECENT (version 1.0.0)[82] using the default parameters without spike-ins. The cutoff for signature genes was fold-change > 2 and false discovery rate (FDR) (Benjamini–Hochberg adjusted $p$ value) <0.01. As the cells in each group came from different samples, we corrected possible batch effects by including a dummy batch variable in the model for the combined data.

**Heatmaps of relative gene expression and signature scores for each cluster**. To calculate the average expression levels of AICDA and MKI67, and the average proliferation scores for each cluster (Supplementary Fig. 10e), a z-score of the normalized expression value was first obtained for every single cell. Then, we calculated the mean z-scores for individual cells in the same cluster and drew them in the heatmap.

To draw the heatmap in Supplementary Fig. 21a and Supplementary Fig. 22b, the average expression levels were calculated for IG genes, IGH genes, IGK genes, IGL genes (Supplementary Fig. 21a), genes in the Kyoto Encyclopedia of Genes and Genomes (KEGG) antigen processing and presentation gene set, and genes in the hallmark interferon-alpha response gene set (Supplementary Fig. 22b) for each single B cell. Then, a z-score of the value was obtained for every cell. The mean z-scores for individual cells in the same cluster in TNBC or PBMC are shown in the heatmap.

**Gene set enrichment analysis**. GSEA was performed using GSEA (version 4.0.3) to identify enriched signatures obtained from the Molecular Signatures Database (MSigDB), including hallmark gene sets, C1 positional gene sets, C2 curated gene sets, C3 motif gene sets, C4 computational gene sets, C5 GO gene sets, C6 oncogenic gene sets, and C7 immunologic gene sets. KEGG gene sets from C2 curated gene sets were selected for GSEA. The complete results are included in Supplementary Data 7.

**LABC and HER2BC B cell annotation**. Scmap (version 1.4.1) was used to map each LABC and HER2BC B cell to annotate TNBC B cell clusters based on expression profiling similarity[83]. Each TNBC patient's B cell feature-barcode matrix, the reference data, was normalized by library size and $\log_2$ transformed by the SingleCellExperiment package. Then, the feature genes most relevant to the underlying biological differences of each B cell cluster in the reference data were defined as the top 2000 residuals of the linear model based on the log(expression) vs. log(dropout) distribution by selectFeatures. The centroid of the 2000 selected genes was calculated for each B cell cluster in the reference data using indexCluster with the default parameters. Finally, the mapping of each B cell expression dataset from LABC and HER2BC to the reference library was carried out by scmapCluster with the default parameters.

**Analysis of TIL-B single-cell RNA-seq data from Wu et al.[53]**. The processed single-cell RNA-seq data and metadata containing cell type information were downloaded from the Gene Expression Omnibus (GEO) database under accession number GSE139555 (File GSE139555_RAW.tar and GSE139555_cd45_nont_metadata.txt.gz). The TIL-B cell numbers were 648 and 904 for the two colon cancer samples; 1421 and 359 for the two lung cancer samples; and 55, 172, and 1832 for the three renal cancer samples. Scmap (version 1.4.1) was used to map those B cells to annotated TNBC B cell clusters based on expression profiling similarity[83], the same method used for the cell type annotation of LABC and HER2BC B cells. The unassigned cell percentages were 0.6% and 3.7% when mapping to the 4 B cell subgroups and 13 B cell subgroups, respectively.

For the subject clustering analysis, the percentage of the B cell subgroup in each sample was calculated first, and then the percentage was scaled across subjects by removing the mean and dividing by the standard deviation. Unassigned B cells from the above scmap analysis were excluded before the percentage calculation. Clustering was conducted by the pheatmap R package using correlation similarity and complete linkage.

**Survival analyses**. RNA-Seq gene expression profiles (Level 3) and clinical data for all tumor types were downloaded from the TCGA data portal (https://gdc-portal.nci.nih.gov/). The METABRIC gene expression profiles and clinical data were downloaded from the cBioPortal website (http://www.cbioportal.org/study?id=brca_metabric#summary). The TNBC samples were defined by IHC results as HER2−, ER−, and PR−, and samples with the normal subtype by PAM50 were removed ($n = 279$). The HER2BC, LABC, and LBBC subtypes were from the PAM50 molecular subtype classifier. The TNBC samples were sorted according to the expression levels of the gene or gene set, and the top 33% ($n = 92$) versus bottom 33% ($n = 92$) of samples were used to generate survival curves. Multivariable analyses were adjusted by lymph node status, tumor size, age at diagnosis, and histological grade. The same samples were used for HR calculation in the Cox regression models, with gene/signature used as categorical variable based on the top 0.33 and bottom 0.33 cut off. For survival analysis and HR calculation, the R package survival (version 2.42–6)[84] was used.

**Paired BCR and single-cell RNA-seq data analyses**. To trace the proliferation and differentiation of different B cell clusters, we counted the same *IGH* germline events and the same *IGH* sequence events within and between each B cell cluster. The same germline event was counted when two cells were in the same clonal B cell tree, and the same *IGH* sequence event was counted when two cells had the same observed *IGH* sequences. The same germline events and the same *IGH* sequence events could be grouped into PBMCs (when two cells were both from PBMC samples), TNBCs (when two cells were both from TNBC samples), or PBMCs/TNBCs (when one cell was from a TNBC sample and the other cell was from a PBMC sample). To compare the levels of the same germline event and the same *IGH* sequence event among different B cell clusters, they were normalized by the formula (2): $N_{normalized} = \frac{N}{N_A} \times \frac{N}{N_B} \times 10,000$ in the heatmap (Fig. 3d, Fig. 4d). $N_{normalized}$ is the normalized value shown in Fig. 3d and Fig. 4d. $N$ is the same germline event number or the same *IGH* sequence event number. $N_A$ and $N_B$ are the cell numbers of the two B cell clusters where the two cells of the same germline or the same *IGH* sequences were from. Then, heatmaps of the same germline or the same *IGH* sequence were generated for PBMCs, TNBCs, or PBMCs/TNBCs using the normalized values.

**Cell cycle analysis**. Cell cycle scores were calculated as the percentages of cell cycle genes' UMIs out of the total UMIs in a cell[25]. We also used the CellCycleScoring function of the Seurat package[40] to calculate the S phase score and G2/M score. The cells were considered to be in the S phase when the S phase score was >0.1 and > the G2/M score. The cells were considered to be in the G2/M phase when the G2/M phase score was >0.1 and > the S phase score.

**Reporting summary**. Further information on research design is available in the Nature Research Reporting Summary linked to this article.

## Data availability

The raw sequence data reported in this paper have been deposited in the Genome Sequence Archive in BIG Data Center, Beijing Institute of Genomics (BIG), Chinese Academy of Sciences, under accession numbers HRA000477 that can be accessed at https://bigd.big.ac.cn/gsa-human/browse/HRA000477. RNA-Seq gene expression profiles (Level 3) and clinical data for all tumor types were downloaded from the TCGA data portal (https://gdc-portal.nci.nih.gov/). The METABRIC gene expression profiles and clinical data were downloaded from the cBioPortal website (http://www.cbioportal.org/study?id=brca_metabric#summary). TIL-B single-cell RNA-seq data from Wu et al.[53] was downloaded from the Gene Expression Omnibus (GEO) database under accession number GSE139555 (File GSE139555_RAW.tar and GSE139555_cd45_nont_metadata.txt.gz). Data for public B cell clone analysis was downloaded from https://github.com/briney/grp_paper. All other data are available in the main text or the Supplementary Information. Source data are provided with this paper.

## Code availability

Custom code used in this study is available at https://github.com/huqingtao2018/B-cells-scRNAseq-in-BC.

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

## Acknowledgements

We thank all breast cancer patients who participated in this study at Xinxiang central hospital. We thank members of Y.Z. lab for helpful discussion and support. The research in Y.Z. the lab is supported by the National Natural Science Foundation of China (81572795, 81773304), the "Hundred, Thousand and Ten Thousand Talent Project" by Beijing municipal government (2019A39). Q.H. is supported by the National Natural Science Foundation of China (31701135). P.Q. is supported by the 2018 Joint Construction Project of Henan Medical Science and Technology Tackling Plan (2018020927). We thank the municipal government of Beijing and the Ministry of Science and Technology of China for funds allocated to NIBS.

## Author contributions

Q.H., Y.H., P.Q., and Y.Z. conceived the study. Q.H., Y.H., P.Q., G.L., X.M., L.G., Z.J., J.W., and T.C. performed experiments and analyzed data. S.X., X.H., Y.G., and P.Q. collected patient samples. Y.Z. analyzed data and wrote the manuscript with support from all authors.

## Competing interests

The authors declare no competing interests.
