## [Peer Review File · Nature Communications]

Reviewers' comments:

Reviewer #1 (Remarks to the Author):

Authors describe single cell sequencing on an impressive number of CD45+ TIL from 10 primary breast cancers with PBMC, the majority of which are TNBC, with a couple of Luminal and one HER2 BC.

They show B cells are really the minority population and focus on B cells, but a vast number of cells are profiled in this work which are not really commented on.

Analysis is largely focused on the B cell population in the TNBC population.

If data are publicly available (both tumor and TIL and TCR/BCR sequences), it will be an excellent resource for other investigators.

Unclear what the main message is, the paper is largely descriptive.

Major

1. Can the make comments on the comparison on the luminal and HER2 B cell populations. Discussion on the heterogeneity in the other immune cells must be made as well.
2. Are these B cells clusters different from the other TNBC? What about the percentage of B cells as well as mutation frequency.
3. Why was exome sequencing preformed? Was there some sort of neoantigen prediction? Did more mutations/antigens correlated with more B/T cells? Title implies such analysis was made. Were any of these BRCA germline mutated?
4. How is a resident memory B cell defined/ distinguished? I believe these are relatively newly defined. It is a big call. So I would urge more caution.
5. Line 291: "Our results also suggested the existence of functionally active germinal centers in TNBC tumor tissues." What supports functionally active?
6. can the prognostic gene signatures be compared to T cell signatures?

Minor

1. odd that the TNBC patients are graded mainly grade 2. Can they give the TIL level of infiltration in Table1A as per previously defined method (www.tilsinbreastcancer.org)
2. Line 117 says 9 pairs of samples were subjected to 5' 10x sequencing. There on only 8 described here. Pls reconcile.
3. Line 282. Pls add reference to support this "which might be due to the heterogeneity of both breast cancer patients and B cell subtypes".
4. Fig S5 is hard to read.

Reviewer #2 (Remarks to the Author):

In this article, Hu Q. et al. aim to better define the molecular characteristics of tumour-infiltrating B cells and plasma cells (TIL-B) in breast cancer. To this end, they used the 10X Genomics platform to obtain (i) single-cell gene expression data and (ii) single-cell T-cell/B-cell receptor sequences (TCR/BCR) from the immune infiltrates (CD45+) and matched peripheral blood mononuclear cells

(PBMCs) of 10 breast cancer patients (n = 6 for triple-negative [TNBC], n = 3 for luminal A and n = 1 for HER2-positive); although most of the paper is focused on 5 TNBC cases. By comparing those two datasets, they argue that, in TNBC, tumor-infiltrating B cells are likely to be involved in tumor control as they show the characteristics of antigen-activated mature and memory B cells and plasma cells. In an attempt to generalize their findings, they used B-cell gene signatures derived from their single-cell gene expression data to perform survival analysis on larger publicly available datasets for breast cancer (METABRIC and TCGA) and other cancer types (TCGA) and argue that such signatures have a better prognostic value than single B-cell markers, in particular CD19 or CD20.

Strengths:

1. Compared to T cells, TIL-B are a vastly understudied immune cell type in the tumour microenvironment; therefore, the subject matter is of interest and likely importance.
2. They applied a state-of-the-art method (scRNA-seq) to a relatively large sample set (by today's standards), and obtained matched transcriptome, TCR and BCR data.

Major issues:

1. The paper lacks any major discoveries. It is well established in breast and other cancers that TIL-B-derived BCRs exhibit hallmarks of antigen recognition, including class switching, somatic hypermutation, and clonal expansion. Thus, the BCR data is largely confirmatory. Their finding of (possibly) 13 molecularly defined subsets of TIL-B is somewhat novel; however, the authors' treatment of this is superficial and does not go beyond what is already known from the literature.
2. In general, the paper lacks adequate discussion and citation of the literature. Although TIL-B are understudied, there are nonetheless dozens of other papers in breast and other cancers that have relevant information yet were not mentioned by the authors.
3. Almost nothing is said about T cell phenotypes, despite this data being available. It would be interesting to see if TIL-B are associated with particular subsets of T cells, such as T_{fh}.
4. In addition to the single-cell sequencing datasets, the authors report whole-exome sequencing data that were used to infer the mutational landscapes of the 6 TNBC cancer cases. Germline and somatic mutations are reported in Table S3, while subsequent analyses on those sets of mutations are reported in Figure S2. However, those two elements are only mentioned in passing (line 129) and add very little to the overall story.
5. The manuscript has many grammatical issues and would benefit from careful editing.

Other issues:

1. In the Methods section, the following points should be addressed:
 - a. Line 365 to 387: Please describe more clearly the procedure to identify and filter germline and somatic mutations. Command lines are useful but require prior knowledge of the software used. It would be helpful to add a sentence explaining briefly what has been done and report the command line in brackets. Detail what are ExAC_ALL and ExAC_EAS, sift score and polyphen score and why such filtering is applied.
 - b. Line 389 to 401: there are no viability markers in the 'Reagents and antibodies' section (starting line 330). How was viability inferred? Trypan blue only? FSC/SSC profile? The gating strategy should be provided as a Supplementary Figure.
 - c. Line 403 to 422: Report code used to filter data. Rather than using 'gene UMI count vs. cell barcode matrix' or the 'UMI cell barcode matrix' (line 426) use the dedicated terminology (in the field: gene expression matrix or UMI count matrix, 10X Genomics: feature-barcode matrix) and provide a brief explanation of what this is.

- d. Line 424 to 440: Provide source code. Is there a reason why the authors decided to focus on the top 10 CCs?
- e. Line 442 to 464: Provide the source code. Did the authors find cells with both BCR and TCR sequences? If so, in what proportion? What are those cells? Doublets that escaped the filtering process? Or real double positive cells?
- f. For the FindCluster function, is it acceptable to arbitrarily set the resolution parameter and compare results? Shouldn't there be an optimal resolution parameter for each dataset? One way to estimate it might be to evaluate how the amount of redundancy between each cluster's marker genes (Table S4 and S5) varies as a function of the resolution parameter. I would expect that over-resolution would result in highly redundant marker genes. This might be worth exploring especially since the authors report very small (and atypical) clusters that could be 'noise' (C9, C12, C11 and C13 on Figure 4).
- g. Line 466 to 486: Provide custom scripts used in this analysis and more details about the process. Why did the authors only focus on the heavy chain (IgH)? What about kappa and lambda? What is the frequency of cells with two productive IgH rearrangement? Are those cells more likely to be doublets than the others or is that a real signal?
- h. Line 502 to 516: descriptions and associated heatmaps are very unclear (Figure 3D and Figure 4D). What is a germline vs V(D)J event? What is plotted against what in those heatmaps? Where are the PBMC/TNBC heatmaps? Since the signal is symmetrical, it might be worthwhile to plot only half of those heatmaps.
- i. Line 518 to 524: why use two methods to quantify cell cycle if filtering is only done using results from the Seurat function? Are those algorithms giving the same status for most of the cells? How were the threshold defined?

2. In the Figure Legends (starting line 536 – add header to this section), the authors should mention the methodological details required to understand each figure panel or at least refer to the associated Methods section. Also, please mention the statistical test used to assess significance. Finally, describe what is represented (e.g., what is represented by the box and whiskers in your box plots?).

3. Line 107-108: the authors argue that TNBC contains more tumor-infiltrating B lymphocytes than other breast cancer types. Given that tumor architecture can drastically vary from one patient to another, is that really fair to normalize by initial tumor weight? Why not normalize by the total number of relevant cells (live tumor + immune cells)? Is it possible to split the 'Non-TNBC' category into the different breast cancer subtypes and perform the adequate statistical test? Of note, given the non-normal distribution of both samples, a Wilcoxon rank sum test (unpaired, two-sided) might be more appropriate than a Student's t test.

4. Line 109 to 116: are the samples submitted to single-cell sequencing part of the 14 samples presented in Figure S1? If yes, please highlight the samples sent to sequencing on this Figure. Are TNBC 1 to 6 the outliers on that box plot?

5. In Figure S2A, what were the rules applied to select the cancer-related genes? How was the analysis presented in Figure S2C performed? There is no mention in the Methods section and no conclusion is drawn from it in the main text.

6. Line 134 to 138: in the 'Reagents and antibodies' section (Methods, starting line 330), the authors mention that they used the antibodies recognizing the following markers: CD45 (immune cells),

CD56 (NK cells), CD11b (monocyte/macrophages), CD3 (T cells), CD20 and CD19 (B cells). Were the sample subjected to single-cell sequencing stained with this antibody cocktail? If so, how concordant are the percentages when comparing the two methodology?

7. Line 138 to 140: The chosen representation makes it very difficult to assess if cells from each sample contribute equally to each cluster. A more quantitative representation would be preferable, e.g. a stacked barplot showing the number of cells per cluster according to their sample of origin.

8. Line 171 to 179: In Figure 2C, shouldn't significance be assessed using a Fisher's exact test on count data rather than a Student's t test on proportions? In Figure 2D, is that really a probability density represented given that most values > 1? Please clarify legend of Figure S2E. No mention of Figure S2F in the main text.

9. To what extent are clonal B cells shared between patients? This could give a hint with regard to which type of antigens might be recognized by TIL-B.

10. Line 222 to 225: Why are the author considering IL-10+ B cells as negative (Figure S7)? What is the reported frequency of regulatory B cells? According to that number, how many cells of this type are to be expected in this dataset? Does that fit with the number of positive cells? Besides, could the author probe other markers of regulatory B cells. [PMID: 28248202] to reinforce their finding? Any hypothesis concerning their absence/non-detection?

11. Line 274 to the end: The Discussion says relatively little about the role of TIL-B in anti-tumor immunity but mostly focusses on what could be done on the remaining datasets.

Reviewer #3 (Remarks to the Author):

Hu et al. used paired BCR-seq and scRNA-seq to profile infiltrating B cells in triple negative breast cancers. The author mentioned that "While it has been well accepted that T cell mediated adaptive cellular immunity plays important roles in immune response for tumors, the roles of B cells in tumor development and therapy, both positive and negative, have been only proposed recently and are still mostly controversial". Their goal is to gain mechanistic insights into the origin and dynamics of tumor infiltrated immune cells, especially B cells, from this study. However, the analysis on both scRNA-seq and B cell repertoire is rudimentary, thus did not provide much new insights on these questions at all.

Major concerns:

1. Fig. S1 had a lot more non-TNBC samples compared to TNBC samples. What if author increase the sample size on TNBC, would they still be able to make the claim that hCD20+ B cells was significantly higher in TNBCs than in other breast cancer subtypes?

2. They reported that of 33,596 B cells where BCRs can be assembled, there are 5,951 and 16,485 B cells containing a single productive IgH allele for BC and PBMC samples. Because of allelic exclusion on IgH, this basically means that 67% B cells are indeed single B cells, while 33% B cells are non-singleton, which is really high. This is really high, especially, after doublet filtering step. They did not offer any explanation or discussion on this at all.

3. From Fig. 3D, they "found that memory B cells contribute mostly for the IgH BCR clonal trees in TNBC samples, suggesting extensive SHM in those infiltrated memory B cells happening inside

tumor.” I am afraid that authors are mixing several concepts here. Contributing to BCR clonal trees does not necessarily mean that they have extensive SHM. They haven’t shown any comparison on SHM in different sub-set of B cells between two anatomical locations, BC and PBMC.

4. Looking at Fig. 3D and Fig. 4D., I am puzzled by the legend. Fig. 3D showed that Bmem and Bmem has the highest overlapping cells. Judging by the legend, this should be around 14380 cells. But the Fig. 4D showed that overlapping between two Bmem sub-sets could reach 28808. This doesn’t make sense. In addition, they indicated earlier that only 22,436 cells are with single productive IgH allele. So, they used other cells, what could have more than one productive allele for it? If the total 33,509 B cells also include cells without IgH but only IgL detected, then what are their criteria for germline comparison?

5. “lasma cells (C11) have the highest percentage of cells with the same IgH allele within clusters in both TNBC and PBMC, suggesting their clonal proliferation.” Do they suggest plasma cells can proliferate or they had been proliferated?

6. They rushed to make a suggestion that CD1chigh B memory cells could be the major direct progenitor cells of these B cells in TNBC based on the observations that they have the most shared the same IgH with other B cell clusters. “the most hared the same IgH” doesn’t make sense.

7. Most of their gene expression analysis, at most, define some B cell subsets. A lot of the functional changes were ignored. Also, they claimed that “most of TNBC infiltrated B cells, except germinal center B cells, (FigS8), are quiescent.” Just using cell cycle gene to classify B cell quiescence is an over simplification.

8. For the rare cell populations, such as C12 germinal center B cells, how many patients did they observe this from? There seems inconsistence between Fig. 4A and Fig. S6 on this population. If they only observe this a subset of patients, then I don’t think they can draw the conclusion that tumor harbor germinal center structure. In addition, this conclusion need additional evidences and supports, which are lacking.

9. They barely discussed C2 naïve B cells and the differences between C1 naïve before jumping to use gene signature to correlate survival. In addition, how do these correlations compare to correlations based on T cell gene signatures?

10. Another problem with this paper is that they did not provide any insights into the origin and dynamics of infiltrated immune cell, especially B cell in tumors or more specifically TNBC, which are what they set out to do.

Other problems:

The paper will be benefited from a language editing. In addition, many of the figures miss detailed figure captions.

Reviewer #1(Remarks to the Author):

Authors describe single cell sequencing on an impressive number of CD45+ TIL from 10 primary breast cancers with PBMC, the majority of which are TNBC, with a couple of Luminal and one HER2 BC.

They show B cells are really the minority population and focus on B cells, but a vast number of cells are profiled in this work which are not really commented on.

Analysis is largely focused on the B cell population in the TNBC population.

If data are publicly available (both tumor and TIL and TCR/BCR sequences), it will be an excellent resource for other investigators.

Unclear what the main message is, the paper is largely descriptive.

Major

1. Can they make comments on the comparison on the luminal and HER2 B cell populations? Discussion on the heterogeneity in the other immune cells must be made as well.

We added more analysis and comments on the luminal and HER2 B TIL-B cells in comparing with TNBC B cells.

In FigS1B, we split the non-TNBC tumor samples into HER2BC, LABC, and LBBC. The TIL-B cell numbers in TNBC were significantly higher than other subtypes.

In FigS5, we demonstrated that the infiltrated B cells in LABC and HER2BC tumors also showed similar BCR (IgH) phenotypes as in TNBC tumors.

The B cell percentages in CD45+ cells for TNBC and non-TNBC tumor samples were similar, ranging from 9% to 51% in TNBC and 5% to 30% in non-TNBC (P value=0.78, Student's two-tailed, t-test) (Fig1C and Table S3).

In FigS13, we added B cell subgrouping analysis results for LABC7-9 and HER2BC10 samples. The B cell clusters annotated in TNBC patients were used as reference and B cells from LABC and HER2BC samples were mapped to the reference based on transcriptional similarity using scmap (Kiselev et al., 2018). Most of the LABC and HER2BC B cells resembled annotated B cell clusters in TNBC (FigS13A,C). Similar as in TNBC, B cells infiltrated in LABC and HER2BC were mostly memory B cells, whereas PBMC samples contained more naïve B cells (FigS13B,D). We also did subject clustering analysis using scaled percentage of 13 B cell subgroups as showed in Figure below, and we did not put the clustering result in the manuscript as more samples were needed to detect stable and clear clustering pattern.

We also checked the naïve B cell and memory B cell signature genes' expression and survival association in other BC subtypes in METABRIC dataset. As expected, higher expression of the naïve B cells and B memory cells gene signatures in TNBC than other BC subtypes was observed, while association with better prognosis was not observed in other BC subtypes (FigS17, TableS10).

In revised manuscript, we discussed the heterogeneity of B cells and other immune cells. “For example, the percentages of B cells in tumor samples varied between 4.6-50.5%, with an average of 22.1%. T cells and NK constituted 21.4-73.7%(average 46.2%) and 0.5-5%(average 2.6%), respectively.”

2. Are these B cells clusters different from the other TNBC? What about the percentage of B cells as well as mutation frequency.

I assume the reviewer was asking similar question about the analysis of non-TNBC samples (LABC and HER2BC) as above.

The infiltrated B cells in non-TNBC tumors also showed similar BCR (IgH) phenotypes, including somatic hypermutation rate and clonality as in TNBC tumors (Fig2 and FigS5).

3. Why was exome sequencing preformed? Was there some sort of neoantigen prediction? Did more mutations/antigens correlated with more B/T cells? Title implies such analysis was made.

Were any of these BRCA germline mutated?

One of the major reasons we performed the exome sequencing was to confirm those tumors were true TNBCs. Originally, as the reviewer mentioned, we would like to perform the neoantigen prediction and correlate it with BCR and/or TCR clones. Unfortunately we could not predict reliable neoantigens reliably so far from our data. More importantly, more works will be required to functionally confirm those neoantigens and correlate them to their true BCRs, which would be far beyond current manuscript. Therefore, we decide to remove the exome sequencing part from the revised manuscript now.

From the exome sequencing results, TNBC5 has a germline splice site mutation in BRCA2 (hg19, chr13:32950930, T>A, HET). No other BRCA mutation was detected in other TNBC patients.

4. How is a resident memory B cell defined/ distinguished? I believe these are relatively newly defined. It is a big call. So I would urge more caution.

Thanks the reviewer to point this out.

We mentioned those tumor-resident memory B cells in line 190-192. Primarily we mean the memory B cells which are resident in the TNBC tumors. Those cells share some characteristics with the recently defined mouse lung resident memory B cells (BRM cells) (Allie et al., Nature Immunology 2019, Palm and Henry, Front Immunol. 2019). For example, they had clonal BCRs that did not significantly overlap with circulating B cells in peripheral blood. They also had higher BCR clonality. On the other hand, in normal breast tissues, there were almost no B cells detected. These results suggested that those TNBC infiltrated memory B cells might develop and differentiate within tumors. However, to completely confirm they are indeed BRM cells, in contrast to circulating memory B cells, further works are necessary. Therefore, we modified the sentence to clarify. We added more discussion on this too in revised manuscript.

5. Line 291: “Our results also suggested the existence of functionally active germinal centers in TNBC tumor tissues.” What supports functionally active?

Those germinal center B cells expressed high level of AICDA, MKI67, and proliferation gene signature (FigS10E).

6. can the prognostic gene signatures be compared to T cell signatures?

We compared our B cell signatures with T cell signatures. As shown in Fig S16, we added CD3G, CD8A, and Trm signature (Savas et al., 2018, Nature Medicine), and compared their HR scores and P values with B cell signatures. The Trm signature showed the best HR scores, followed by Bmem, B naïve, and CD8A. P values of CD3, CD19, and CD20 were not consistently significant among different analysis (OS/DFR, Univirable /Multivirable).

Minor

1. odd that the TNBC patients are graded mainly grade 2. Can they give the TIL level of infiltration in Table1A as per previously defined method (www.tilsinbreastcancer.org)

The H&E sections were double-checked by other pathologists and were confirmed that those TNBC tumors were indeed mainly grade 2.

We used the method reviewer recommended to calculate the TIL levels. The results were added in Fig 1A.

2. Line 117 says 9 pairs of samples were subjected to 5' 10x sequencing. There on only 8 described here. Pls reconcile.

We corrected this mistake. In line 117, "TNBC2-5" should be "TNBC2-6".

3. Line 282. Pls add reference to support this "which might be due to the heterogeneity of both breast cancer patients and B cell subtypes".

We added references in revised manuscript.

4. Fig S5 is hard to read.

We modified the FigS5 and added more details in figure legend. In the revised manuscript, FigS5 was moved to FigS8

Reference

Allie, S.R., Bradley, J.E., Mudunuru, U., Schultz, M.D., Graf, B.A., Lund, F.E., and Randall, T.D. (2019). The establishment of resident memory B cells in the lung requires local antigen encounter. *Nature immunology* 20, 97-108.

Kiselev, V.Y., Yiu, A., and Hemberg, M. (2018). scmap: projection of single-cell RNA-seq data across data sets. *Nature methods* 15, 359-362.

Palm AE, Henry C. Remembrance of Things Past: Long-Term B Cell Memory After Infection and Vaccination. *Front Immunol.* 2019 Jul 31;10:1787.

Savas, P., Virassamy, B., Ye, C., Salim, A., Mintoff, C.P., Caramia, F., Salgado, R., Byrne, D.J., Teo, Z.L., Dushyanthen, S., et al. (2018). Single-cell profiling of breast cancer T cells reveals a tissue-resident memory subset associated with improved prognosis. *Nature medicine* 24, 986-993.

Reviewer #2 (Remarks to the Author):

In this article, Hu Q. et al. aim to better define the molecular characteristics of tumour-infiltrating B cells and plasma cells (TIL-B) in breast cancer. To this end, they used the 10X Genomics platform to obtain (i) single-cell gene expression data and (ii) single-cell T-cell/B-cell receptor sequences (TCR/BCR) from the immune infiltrates (CD45+) and matched peripheral blood mononuclear cells (PBMCs) of 10 breast cancer patients (n = 6 for triple-negative [TNBC], n = 3 for laminal A and n = 1 for HER2-positive); although most of the paper is focused on 5 TNBC cases. By comparing those two datasets, they argue that, in TNBC, tumor-infiltrating B cells are likely to be involved in tumor control as they show the characteristics of antigen-activated mature and memory B cells and plasma cells. In an attempt to generalize their findings, they used B-cell gene signatures derived from their single-cell gene expression data to perform survival analysis on larger publicly available datasets for breast cancer (METABRIC and TCGA) and other cancer types (TCGA) and argue that such signatures have a better prognostic value than single B-cell markers, in particular CD19 or CD20.

Strengths:

1. Compared to T cells, TIL-B are a vastly understudied immune cell type in the tumour microenvironment; therefore, the subject matter is of interest and likely importance.
2. They applied a state-of-the-art method (scRNA-seq) to a relatively large sample set (by today's standards), and obtained matched transcriptome, TCR and BCR data.

Major issues:

1. The paper lacks any major discoveries. It is well established in breast and other cancers that TIL-B-derived BCRs exhibit hallmarks of antigen recognition, including class switching, somatic hypermutation, and clonal expansion. Thus, the BCR data is largely confirmatory. Their finding of (possibly) 13 molecularly defined subsets of TIL-B is somewhat novel; however, the authors' treatment of this is superficial and does not go beyond what is already known from the literature.

We partially agree with reviewer's comments. Honestly speaking, we are very confused with the "major discoveries" the reviewer is expecting. Indeed, most of, if not all, current sc-seq analyses on human tumor samples are largely descriptive and confirmative, such as the several studies on TIL-Ts in various human cancers, which just published in major journals in last a few years. But, here we performed the first comprehensive single-cell analysis on TIL-B, which provides a foundation for further studies on B cell tumor immunology, which would be broadly appreciated.

In the revised manuscripts, following reviewers' suggestions, we have done much deeper analysis and included more discussions for the potential origin and functions of TIL-Bs.

2. In general, the paper lacks adequate discussion and citation of the literature. Although TIL-B are understudied, there are nonetheless dozens of other papers in breast and other cancers that have relevant information yet were not mentioned by the authors.

Thanks reviewer's suggestion! We now included more discussions and cited more previous TIL-B studies.

3. Almost nothing is said about T cell phenotypes, despite this data being available. It would be interesting to see if TIL-B are associated with particular subsets of T cells, such as Tfh.

We analyzed the T cell phenotypes but so far could not find significant correlations of particular subsets of T cells (including Tfh) with TIL-B, which likely due to the limited sample numbers in current study. Therefore we did not include those analyses in the current manuscript.

4. In addition to the single-cell sequencing datasets, the authors report whole-exome sequencing data that were used to infer the mutational landscapes of the 6 TNBC cancer cases. Germline and somatic mutations are reported in Table S3, while subsequent analyses on those sets of mutations are reported in Figure S2. However, those two elements are only mentioned in passing (line 129) and add very little to the overall story.

We removed the whole-exome sequencing part from the revised manuscript.

5. The manuscript has many grammatical issues and would benefit from careful editing.

Now we carefully revised the whole manuscript.

Other issues:

1. In the Methods section, the following points should be addressed:
 - a. Line 365 to 387: Please describe more clearly the procedure to identify and filter germline and somatic mutations. Command lines are useful but require prior knowledge of the software used. It would be helpful to add a sentence explaining briefly what has been done and report the command line in brackets. Detail what are ExAC_ALL and ExAC_EAS, sift score and polyphen score and why such filtering is applied.
 - b. Line 389 to 401: there are no viability markers in the 'Reagents and antibodies' section (starting line 330). How was viability inferred? Trypan blue only? FSC/SSC profile? The gating strategy should be provided as a Supplementary Figure.
 - c. Line 403 to 422: Report code used to filter data. Rather than using 'gene UMI count vs. cell barcode matrix' or the 'UMI cell barcode matrix' (line 426) use the dedicated terminology (in the field: gene expression matrix or UMI count matrix, 10X Genomics: feature-barcode matrix) and provide a brief explanation of what this is.

- d. Line 424 to 440: Provide source code. Is there a reason why the authors decided to focus on the top 10 CCs?
- e. Line 442 to 464: Provide the source code. Did the authors find cells with both BCR and TCR sequences? If so, in what proportion? What are those cells? Doublets that escaped the filtering process? Or real double positive cells?
- f. For the FindCluster function, is it acceptable to arbitrarily set the resolution parameter and compare results? Shouldn't there be an optimal resolution parameter for each dataset? One way to estimate it might be to evaluate how the amount of redundancy between each cluster's marker genes (Table S4 and S5) varies as a function of the resolution parameter. I would expect that over-resolution would result in highly redundant marker genes. This might be worth exploring especially since the authors report very small (and atypical) clusters that could be 'noise' (C9, C12, C11 and C13 on Figure 4).
- g. Line 466 to 486: Provide custom scripts used in this analysis and more details about the process. Why did the authors only focus on the heavy chain (IgH)? What about kappa and lambda? What is the frequency of cells with two productive IgH rearrangement? Are those cells more likely to be doublets than the others or is that a real signal?
- h. Line 502 to 516: descriptions and associated heatmaps are very unclear (Figure 3D and Figure 4D). What is a germline vs V(D)J event? What is plotted against what in those heatmaps? Where are the PBMC/TNBC heatmaps? Since the signal is symmetrical, it might be worthwhile to plot only half of those heatmaps.
- i. Line 518 to 524: why use two methods to quantify cell cycle if filtering is only done using results from the Seurat function? Are those algorithms giving the same status for most of the cells? How were the threshold defined?

We revised the method section according to reviewer's comments:

- a. We removed the whole-exome sequencing part.
- b. We added the gating strategy in Fig S1A. We used FSC/SSC and also DAPI staining to exclude dead cells. We added this in the method section.
- c. We modified the method part according to reviewer's suggestions. We used the dedicated terminology 'feature-barcode matrix', and a brief explanation 'the number of UMIs associated with a feature (row) and a barcode (column)' in the revised manuscript. All of the source codes needed to repeat the cell filtering, grouping, marker gene define and BCR related analyses can be downloaded from the GitHub (<https://github.com/huqingtao2018/B-cells-scRNAseq-in-BC>).
- d. We add the source code in the GitHub (<https://github.com/huqingtao2018/B-cells-scRNAseq-in-BC>). The CC number was determined by inspecting the results of DimHeatmap of Seurat package. Then we use CalcVarExpRatio to calculate the

percentage of variance explained by the 10 CCs for each cell. There were 94.24% cells with 50% or more variance explained when CC10 was used. We included this in the revised method part.

- e. We add the source code in the GitHub (<https://github.com/huqingtao2018/B-cells-scRNAseq-in-BC>). 1,195 BCR and TCR double positive cells were detected, which were 4.5% of all BCR positive cells or 2.7% of all TCR positive cells. They had significantly more UMIs than BCR single-positive cells, TCR single-positive cells, and BCR-negative/TCR-negative cells, suggesting they were B cell and T cell doublets (FigS3A, FigS3C).
- f. The resolution parameter was not set arbitrarily. We first projected canonical B cell related markers (CD20, IGHD, CD27, and CD38) onto tSNE plot to visualize the expression pattern. Naïve B cell (CD20⁺CD38⁻CD27⁻IGD⁺), memory B cells (CD20⁺CD38⁻CD27⁺), and plasma B cells (CD20⁺CD38⁺) could be well separated in the tSNE plot (FigS6). Parameter res0.1 of FindCluster function can cluster cells into 4 groups, and 3 of them fit the marker projection of naïve, memory, and plasma B cells in the tSNE plot. The other group is marked by CD14⁺ expression. Then we made a general description of the B cells under parameter res0.1 in Figure3. In addition, switched memory B cells and non-switched memory B cells could be well separated (FigS7B).

To define an optimal resolution parameter is challenging for unsupervised clustering method of single-cell RNA-seq data. There is no consensus on the correct methods for selecting the resolution of clustering (Kiselev VY et al., 2019, Nat Rev Genet). As we were interested in uncovering more B cell subtypes in this dataset, we increased the resolution to 0.6, 0.8, and 1 and checked the grouping results. In resolution 1, GC B cell (CD20⁺BCL6⁺) group with only 43 cells could be well clustered. For the GC B cell group, they were detected in 4 of the 5 TNBC tissues, and AICDA was highly expressed there, indicating they were functionally active GC B cells (FigS10E). In addition, we observed Tfh cell populations in the same TNBC samples (data not shown). We also observed TLS structures by CD3 and CD20 IHC in all 6 TNBC patients (FigS10F). All of these results suggested this small group of GC B cells was not 'noise' in our dataset and parameter resolution 1 could give reliable clustering result for this small group. Then we checked the marker genes for each of the 13 groups under this resolution in order to define more B cell groups. We could annotate all of them according to marker gene expression patterns in Fig4B. We add more details in the method part for resolution selection.

We also tried to evaluate how the amount of redundancy between each cluster's marker genes varied as a function of the resolution parameter (From resolution 0.1 to 3 by a step of 0.1). As expected by the reviewer, we could see that at resolution 0.1 the relative numbers of redundant marker gene to total numbers of marker gene were the lowest and at res1 it's around 25%.

g. We added more details in revised manuscript method part and added the code in the GitHub (<https://github.com/huqingtao2018/B-cells-scRNAseq-in-BC>). Contigs of IgH, IgK, and IgL were assembled from the BCR libraries. But IgK and IgL contig assembling rates were significant lower than IgH contigs (data not shown). Since the diversity of IGH is sufficient for most antibody specificities, we only used sequences of IgH to include more B cells into further analysis (Xu JL et al., 2000, Immunity). Cells with multiple productive IgHs were detected (~14%) and removed before further analysis. They had significantly more UMIs than single productive IgH B cells, suggesting that they were B cell doublets (FigS3A and FigS3B).

h. We are very sorry we did not clearly describe the heatmaps. We updated the method part for ‘Paired BCR and single-cell RNA sequencing data analyses’ with more details in the revised manuscript. We use ‘same IgH’ in the revised manuscript, instead of same V(D)J event (an example figure was showed below).

For a same germline event between two B cell clusters, it means there were two clonally related B cells (with the same germline), and one cell was from B cell clusters in rows and the other cell was from B cell clusters in columns of the heatmap.

To plot Fig3D and Fig4D, the same germline events between two B cell clusters were calculated using all clonally related B cells belonging to these two B cell clusters. First, one same germline event was counted for the two B cell clusters for any two clonally related B cells. Then the sum of the same germline events between two B cell clusters was normalized by dividing cell numbers of these two clusters in order to compare among those B cell clusters with different cell numbers. The normalization method was

$$N_{\text{normalized}} = \frac{N}{N_A} \times \frac{N}{N_B} \times 10,000$$

where $N_{\text{normalized}}$ is the normalized value showed in

Fig3D and Fig4D; N is the same germline event numbers between cell cluster A and cell cluster B; N_A and N_B are the cell numbers of the cell cluster A and B.

The same germline event could be grouped into PBMC (when two cells were both from PBMC samples), TNBC (when two cells were both from TNBC samples), or across PBMC/TNBC (when one cell was from TNBC samples and the other cell was from PBMC samples). The same IgH heatmap was drew in the same way as the same germline,

except the same IgH numbers were counted only when the two clonally related B cells have exactly the same observed IgH sequence, instead of sharing the same germline sequence. We did not show the PBMC/TNBC heatmaps in the manuscript, since the event numbers were low.

The signal was symmetrical. We plotted the whole heatmap in the manuscript to show the diagonal values more clearly, which represented intra-cluster same germline and intra-cluster same IgH event numbers.

An example of how same germline and same IgH sequence event were calculated and normalized in the heatmaps of Fig3D and Fig4D. The left panel were clonal related B cells in lineage tree format. Germline at the top represents germline B cell of this lineage tree. Black dot represent inferred cells. Blue color represents B cells from PBMC and red color represents B cells from TNBC. Larger circle represents larger cell numbers. Text in circle represents B cell cluster ID for the B cells. Number on line represents mutation number between the B cells at each end of the line. N1, N2 and N3 were cell numbers in C1, C2 and C3.

- i. We used two different algorithms to evaluate the proliferation state to make the result more reliable. The first algorithm used 53 cell cycle related genes to calculate the proliferation score and it reported one proliferation score for each cell (Li H et al., 2018, Cell). The cutoff of proliferating was defined by bimodal distribution as described in Li et al., 2018 paper. The second algorithm used 43 G1/S related genes and 54 G2/M related genes in Seurat package to calculate proliferation score. It reported two scores (g1 score and g2m score) and cell cycle phase (G1, S, or G2M) for each cell. The cutoff of phase assign was modified from previous report (Rodda LB et al., 2018, Immunity). When the G1/S score ≤ 0.1 and G2/M score ≤ 0.1 , it was defined as G1 phase. When the G1/S score > 0.1 and $> G2/M$ score, it was defined as S phase. When the G2/M score > 0.1 and $> G1/S$ score, it was defined as G2/M phase. As shown in FigS15, the two algorithms showed similar results that C12 GC B cell was actively proliferating. The following figure showed the distribution of proliferation scores from the first algorithm for cells with different cell cycle phase by the second algorithm. G2M cell state by the second algorithm showed the higher proliferation score by the first algorithm. For G1 phase and S phase, the two algorithms did not fit well. Two algorithms got similar results for G2M phase cells, but not for G1 and S phase cells.

2. In the Figure Legends (starting line 536 - add header to this section), the authors should mention the methodological details required to understand each figure panel or at least refer to the associated Methods section. Also, please mention the statistical test used to assess significance. Finally, describe what is represented (e.g., what is represented by the box and whiskers in your box plots?).

We added details as reviewer suggested in the revised manuscript.

3. Line 107-108: the authors argue that TNBC contains more tumor-infiltrating B lymphocytes than other breast cancer types. Given that tumor architecture can drastically vary from one patient to another, is that really fair to normalize by initial tumor weight? Why not normalize by the total number of relevant cells (live tumor + immune cells)? Is it possible to split the 'Non-TNBC' category into the different breast cancer subtypes and perform the adequate statistical test? Of note, given the non-normal distribution of both samples, a Wilcoxon rank sum test (unpaired, two-sided) might be more appropriate than a Student's t test.

We used Lymphoprep to enrich the lymphocytes before FACS analysis. Therefore we have to normalize by tumor weight, which has been also frequently used by others.

We split the non-TNBC tumor samples and used Wilcoxon rank sum test as suggested in revised FigS1B.

4. Line 109 to 116: are the samples submitted to single-cell sequencing part of the 14 samples presented in Figure S1? If yes, please highlight the samples sent to sequencing on this Figure. Are TNBC 1 to 6 the outliers on that box plot?

We highlighted those samples in FigS1 as suggested. TNBC1-6 were not outliers on the box plot.

5. In Figure S2A, what were the rules applied to select the cancer-related genes? How was the analysis presented in Figure S2C performed? There is no mention in the Methods section and no conclusion is drawn from it in the main text.

We deleted the whole-exome sequencing part in the revised version.

6. Line 134 to 138: in the ‘Reagents and antibodies’ section (Methods, starting line 330), the authors mention that they used the antibodies recognizing the following markers: CD45 (immune cells), CD56 (NK cells), CD11b (monocyte/macrophages), CD3 (T cells), CD20 and CD19 (B cells). Were the sample subjected to single-cell sequencing stained with this antibody cocktail? If so, how concordant are the percentages when comparing the two methodology?

The FACS antibodies for CD45 (immune cells), CD3 (T cells), CD20 and CD19 (B cells) were used for the samples subjected to single-cell sequencing. The antibodies for CD56 and CD11b were used only for some test experiment samples. We removed CD56, CD11b FACS antibodies from “reagents and antibodies” section.

We compared results obtained by FACS and single-cell sequencing for T cell group, B cell group and the rest cell as one group. Overall, the correlations between two methodologies were good for T cell, B cell, and the rest cell (spearman correlation r values were 0.59, 0.59, and 0.38 respectively). T cell frequencies were significantly higher in scRNA-Seq than in flow cytometry (Student’s two-tailed, paired t-test, $P=0.00046$).

7. Line 138 to 140: The chosen representation makes it very difficult to assess if cells from each sample contribute equally to each cluster. A more quantitative representation would be preferable, e.g. a stacked barplot showing the number of cells per cluster according to their sample of origin.

We added the stacked barplots as reviewer suggested in Fig S2C, FigS7C, and FigS10C.

8. Line 171 to 179: In Figure 2C, shouldn't significance be assessed using a Fisher's exact test on count data rather than a Student's t test on proportions? In Figure 2D, is that really a probability density represented given that most values > 1? Please clarify legend of Figure S2E. No mention of Figure S2F in the main text.

For Figure 2C, we used Fisher's exact test on count data but we did not describe it in the legend. Now we have updated it in the revised version.

Fig2D is a smoothed version of the histogram by kernel density estimate. Histograms before smoothing were shown below. There are 140 and 190 clones in TNBC and PBMC tissue respectively (73 clones with both TNBC and PBMC cells were not included in this analysis). The mutation frequency of a clone was calculated as the average of mutation frequencies of all sequences in that clone. We added more details in figure legends.

We added more description to legend of FigS2E. Figure S2F showed example clonal trees with TNBC cells only, PBMC cells only and both TNBC and PBMC cells. We mentioned this figure in the revised manuscript. In the revised manuscript, FigS2 was moved to FigS4.

9. To what extent are clonal B cells shared between patients? This could give a hint with regard to which type of antigens might be recognized by TIL-B.

In our data, none of the clonal B cells were shared between patients.

10. Line 222 to 225: Why are the author considering IL-10+ B cells as negative (Figure S7)? What is the reported frequency of regulatory B cells? According to that number, how many cells of this type are to be expected in this dataset? Does that fit with the number of positive cells? Besides, could the author probe other markers of regulatory B cells. [PMID: 28248202] to reinforce their finding? Any hypothesis concerning their absence/non-detection?

While numbers of publications have suggested different markers for Bregs (such as IL-10–producing CD24^{hi}CD38^{hi} B cells, CD24^{hi}CD27⁺ B cells (B10), CD38⁺CD1d⁺IgM⁺CD147⁺GrB⁺ B cells, CD27^{int}CD38^{hi} plasmablasts, and CD19⁺TIM1⁺ B cells), IL-10 is the only one accepted widely in the field without controversy. It also is proposed that Bregs regulate other immune cells mainly by secretion of IL-10.

The estimated percentages of Bregs have been reported in HCC (10% of all B cells, Xiao X et al., 2016, Cancer Discovery), in lung cancer (11% of B cells, Lizotte PH et al., 2016, JCI Insight), and in gastric cancer (~10% of B cells, Wang WW et al., 2015 Oncotarget). If breast cancers have the same percentage of Breg as those types cancers (around 10%), we would expect to observe a Breg cluster with ~863 cells (out of 8,632 B cells in TNBC tissues and PBMC tissues) or ~252 cells (out of 2,526 B cells in TNBC tissues). However, we could not detect them in our data. The smallest TIL-B subgroup (germinal center B cells, C12) we could well detected in current studies was about 0.5% of total TIL-Bs.

By analyzing macrophages and monocytes that also express IL-10 (FigS12), we ruled out the possibility that technical limitations of single cell sequencing might lead to inefficient detection of IL-10 expression in B cells. In addition, those a few IL-10 expressing B cells could not form distinct cluster separating from the rest of B cells. Moreover, we could not find negative association between the gene expression signatures of any B cell subgroups with prognosis of TNBC patients. Therefore, we are confident that IL-10 expressing B cells (Bregs) were absence in our TNBC samples.

We also checked GZMB and PD1, other potential Breg markers. Their expressions could be well detected in NK cells and/or T cells, but we could not identify specific GZMB- or PD1-expressing B cell populations (FigS12)

The presence of Bregs might be tumor specific and/or stage specific. However, such hypothesis is too preliminary to be discussed in current manuscript.

11.Line 274 to the end: The Discussion says relatively little about the role of TIL-B in anti-tumor immunity but mostly focusses on what could be done on the remaining datasets.

As the reviewer suggested, we included more discussions about the roles of TIL-B in anti-tumor immunity. We believe that the whole large dataset we generated in this study may benefit others significantly.

References

Kiselev VY, Andrews TS, Hemberg M. Challenges in unsupervised clustering of single-cell RNA-seq data. Nat Rev Genet. 2019 May;20(5):273-282.

Li H, van der Leun AM, Yofe I, Lubling Y, Gelbard-Solodkin D, van Akkooi ACJ, van den Braber M, Rozeman EA, Haanen JBAG, Blank CU, Horlings HM, David E, Baran Y, Bercovich

A, Lifshitz A, Schumacher TN, Tanay A, Amit I. Dysfunctional CD8 T Cells Form a Proliferative, Dynamically Regulated Compartment within Human Melanoma. *Cell*. 2019 Feb 7;176(4):775-789.e18.

Lizotte PH, Ivanova EV, Awad MM, Jones RE, Keogh L, Liu H, Dries R, Almonte C, Herter-Sprie GS, Santos A, Feeney NB, Paweletz CP, Kulkarni MM, Bass AJ, Rustgi AK, Yuan GC, Kufe DW, Jänne PA, Hammerman PS, Sholl LM, Hodi FS, Richards WG, Bueno R, English JM, Bittinger MA, Wong KK. Multiparametric profiling of non-small-cell lung cancers reveals distinct immunophenotypes. *JCI Insight*. 2016 Sep 8;1(14):e89014.

Rodda LB, Lu E, Bennett ML, Sokol CL, Wang X, Luther SA, Barres BA, Luster AD, Ye CJ, Cyster JG. Single-Cell RNA Sequencing of Lymph Node Stromal Cells Reveals Niche-Associated Heterogeneity. *Immunity*. 2018 May 15;48(5):1014-1028.e6

Wang WW, Yuan XL, Chen H, Xie GH, Ma YH, Zheng YX, Zhou YL, Shen LS. CD19+CD24hiCD38hiBregs involved in downregulate helper T cells and upregulate regulatory T cells in gastric cancer. *Oncotarget*. 2015 Oct 20;6(32):33486-99.

Xiao X, Lao XM, Chen MM, Liu RX, Wei Y, Ouyang FZ, Chen DP, Zhao XY, Zhao Q, Li XF, Liu CL, Zheng L, Kuang DM. PD-1hi Identifies a Novel Regulatory B-cell Population in Human Hepatoma That Promotes Disease Progression. *Cancer Discov*. 2016 May;6(5):546-59.

Xu JL, Davis MM. Diversity in the CDR3 region of V(H) is sufficient for most antibody specificities. *Immunity*. 2000 Jul;13(1):37-45.

Reviewer #3 (Remarks to the Author):

Hu et al. used paired BCR-seq and scRNA-seq to profile infiltrating B cells in triple negative breast cancers. The author mentioned that “While it has been well accepted that T cell mediated adaptive cellular immunity plays important roles in immune response for tumors, the roles of B cells in tumor development and therapy, both positive and negative, have been only proposed recently and are still mostly controversial”. Their goal is to gain mechanistic insights into the origin and dynamics of tumor infiltrated immune cells, especially B cells, from this study. However, the analysis on both scRNA-seq and B cell repertoire is rudimentary, thus did not provide much new insights on these questions at all.

Major concerns:

1. Fig. S1 had a lot more non-TNBC samples compared to TNBC samples. What if author increase the sample size on TNBC, would they still be able to make the claim that hCD20+ B cells was significantly higher in TNBCs than in other breast cancer subtypes?

We split the non-TNBC samples to different subtypes. As shown in FigS1B, the TNBC samples still had significantly more CD20+ cells than luminal and HER2 BCs. This result is in consistent with published results (Cimino-Mathews A et al., 2016, Hum Pathol; Mahmoud SM et al., 2012, Breast Cancer Res Treat). The TNBC samples are more difficult to be collected since only 20% or less BCs are TNBC. It will take us at least several months to increase the sample size on TNBC for this confirmative experiment.

2. They reported that of 33,596 B cells where BCRs can be assembled, there are 5,951 and 16,485 B cells containing a single productive IgH allele for BC and PBMC samples. Because of allelic exclusion on IgH, this basically means that 67% B cells are indeed single B cells, while 33% B cells are non-singleton, which is really high. This is really high, especially, after doublet filtering step. They did not offer any explanation or discussion on this at all.

We are very sorry that we did not clearly state all cell numbers after various filtering steps in previous version. Now we added a summary in FigS3A and more detailed descriptions in method part. There were 33,509 cells with UMI \geq 2 in BCR libraries. The sequencing reads for those cells were assembled into contigs by Cell Ranger VDJ pipeline and then were annotated by mapping to IMGT database. 26,401 of them contained IgH contigs. Among them, there were 26,172 cells with productive IgH, and 22,436 (86%) of them had single productive IgH. For the 14% multiple productive IgH cells, significantly more UMIs were detected (FigS3B), indicating they were doublets. We removed them before further analysis.

3. From Fig. 3D, they “found that memory B cells contribute mostly for the IgH BCR

clonal trees in TNBC samples, suggesting extensive SHM in those infiltrated memory B cells happening inside tumor.” I am afraid that authors are mixing several concepts here. Contributing to BCR clonal trees does not necessarily mean that they have extensive SHM. They haven’t shown any comparison on SHM in different sub-set of B cells between two anatomical locations, BC and PBMC.

Thank reviewer for pointing this out and we agreed with reviewer’s comments. We modified this in the revised manuscript. We added the comparisons as reviewer suggested in FigS9. Indeed, both in PBMC and TNBC samples, SHM frequencies of Bmem and Plasma B cell were significantly higher than that of naïve B cells. Overall, SHMs of B cells in TNBC were higher than in PBMC, except that plasma cells had higher SHM in PBMC than in TNBC. We add this into the revised manuscript (FigS9).

4.Looking at Fig. 3D and Fig. 4D., I am puzzled by the legend. Fig. 3D showed that Bmem and Bmem has the highest overlapping cells. Judging by the legend, this should be around 14380 cells. But the Fig. 4D showed that overlapping between two Bmem sub-sets could reach 28808. This doesn’t make sense. In addition, they indicated earlier that only 22,436 cells are with single productive IgH allele. So, they used other cells, what could have more than one productive allele for it? If the total 33,509 B cells also include cells without IgH but only IgL detected, then what are their criteria for germline comparison?

We are very sorry that we did not clearly describe the heatmaps. The legend represented same germline event numbers or same IgH sequence event numbers normalized by cell numbers of cell groups in the rows and columns of the heatmaps (an example figure was showed below). As we used a constant value of 10,000 for normalization, the numbers in the heatmaps could be larger than cell numbers sometimes. We clarified the confusion for how we calculated the numbers and how we did the normalization in the revised method part for ‘Paired BCR and single-cell RNA sequencing data analyses’. We only analyzed single productive IgH B cells, and we removed B cells without IgH but had only IgL detected before analysis.

An example of how same germline and same IgH sequence event were calculated and normalized in the heatmaps of Fig3D and Fig4D. The left panel were clonal related B cells in lineage tree format. Germline at the top represents germline B cell of this lineage tree. Black dot represent inferred cells. Blue color represents B cells from PBMC and red color represents B cells from TNBC. Larger circle represents larger cell numbers. Text in circle represents B cell cluster ID for the B cells. Number on line represents mutation number between the B cells at each end of the line. N1, N2 and N3 were cell numbers in C1, C2 and C3.

5. “plasma cells (C11) have the highest percentage of cells with the same IgH allele within clusters in both TNBC and PBMC, suggesting their clonal proliferation.” Do they suggest plasma cells can proliferate or they had been proliferated?

The plasma cells showed the lowest proliferation scores among all B cell groups (FigS10E), suggesting that they had been proliferated but not were proliferating.

6. They rushed to make a suggestion that CD1chigh B memory cells could be the major direct progenitor cells of these B cells in TNBC based on the observations that they have the most shared the same IgH with other B cell clusters. “the most hared the same IgH” doesn’ t make sense.

We removed this speculation/hypothesis in revised manuscript.

7. Most of their gene expression analysis, at most, define some B cell subsets. A lot of the functional changes were ignored. Also, they claimed that “most of TNBC infiltrated B cells, except germinal center B cells, (FigS8), are quiescent.” Just using cell cycle gene to classify B cell quiescence is an over simplification.

We performed deeper analyses and added more discussions in revised manuscript.

We changed the statement in the revised manuscript as ‘On the contrary, most of TNBC infiltrated B cells, except germinal center B cells (FigS15), were not actively proliferating.’

8. For the rare cell populations, such as C12 germinal center B cells, how many patients did they observe this from? There seems inconsistence between Fig. 4A and Fig. S6 on this population. If they only observe this a subset of patients, then I don’ t think they can draw the conclusion that tumor harbor germinal center structure. In addition, this conclusion need additional evidences and supports, which are lacking.

We observed the C12 germinal center B cells in 4 of 5 TNBC samples (FigS10C and TableS8 in the revised manuscript). Those germinal center B cells expressed AICDA and proliferation marker genes significantly (FigS10E). In addition, we also observed Tfh cell populations in the same TNBC samples (data not shown). All of these results are suggesting the existence of the potential tertiary lymphoid structures containing germinal center B cells in TNBC, which have been observed in various tumors including TNBCs (Colbeck et al., 2017, *Frontiers in immunology*; Dieu-Nosjean et al., 2014, *Trends in immunology*). We also observed TLS structures by Immunohistochemistry (IHC) using CD3 and CD20 in all of the 6 TNBC patients (an example IHC result was shown in FigS10F).

For the inconsistence, we add detailed information for Fig4A in FigS10 in the revised version of manuscript. FigS6 contains the detailed information for Fig3. In the revised manuscript, FigS6 was moved to FigS7

Three articles published in Nature recently (Petitprez F et al., 2020, *Nature*; Cabrita R et al.,

2020 Nature; Helmink BA et al., 2020, Nature) demonstrated that B cells and tertiary lymphoid structures could be associated with better outcomes when individuals undergo immunotherapy. Our results provide experimental tools and cellular and molecular mechanisms to understand the roles of B cells and TLS in cancer.

9. They barely discussed C2 naïve B cells and the differences between C1 naïve before jumping to use gene signature to correlate survival. In addition, how do these correlations compare to correlations based on T cell gene signatures?

Thanks for the reviewer to point this out. We compared the expression level of genes between C1 naïve B cells and C2 naïve B cells. C1 cells were Nur77 (NR4A1) positive, and they had BCR activated (marked by Fos/Jun expression), suggesting that they might be anergic naïve B cells which are similar as anergic T cells. On the other hand, C2 cells were classic naïve cells (Zikherman J et al., 2012, Nature; Tan C et al, 2019, J Immunol). The expression of top20 differentially expressed genes (ranked by Fold Change) were selected and showed as heatmap below.

As the reviewer required, we compared our B cell signatures with T cell signatures. As shown in Fig S16, we added CD3G, CD8A, and Trm signature(Savas et al., 2018, Nature medicine), and compared their HR scores and P values with B cell signatures. The Trm signature showed the best HR scores, followed by Bmem, B naïve, and CD8A. P values of CD3, CD19, and CD20 were not consistently significant among different analysis (OS/DFR, Univariable/Multivariable).

10. Another problem with this paper is that they did not provide any insights into the origin and dynamics of infiltrated immune cell, especially B cell in tumors or more specifically TNBC, which are what they set out to do.

We performed deeper analyses and added more discussions in revised manuscript.

Other problems:

1. The paper will be benefited from a language editing. In addition, many of the figures miss detailed figure captions.

Now we carefully revised the whole manuscript and also added more details for figures and legends.

References

Cabrita R, Lauss M, Sanna A, Donia M, Skaarup Larsen M, Mitra S, Johansson I, Phung B, Harbst K, Vallon-Christersson J, van Schoiack A, Lövgren K, Warren S, Jirstrom K, Olsson H, Pietras K, Ingvar C, Isaksson K, Schadendorf D, Schmidt H, Bastholt L, Carneiro A, Wargo JA, Svane IM, Jönsson G. Tertiary lymphoid structures improve immunotherapy and survival in melanoma. *Nature*. 2020 Jan;577(7791):561-565.

Cimino-Mathews A, Thompson E, Taube JM, Ye X, Lu Y, Meeker A, Xu H, Sharma R, Lecksell K, Cornish TC, Cuka N, Argani P, Emens LA. PD-L1 (B7-H1) expression and the immune tumor microenvironment in primary and metastatic breast carcinomas. *Hum Pathol*. 2016 Jan;47(1):52-63.

Colbeck EJ, Ager A, Gallimore A, Jones GW. Tertiary Lymphoid Structures in Cancer: Drivers of Antitumor Immunity, Immunosuppression, or Bystander Sentinels in Disease? *Front Immunol*. 2017 Dec 19;8:1830.

Dieu-Nosjean MC, Goc J, Giraldo NA, Sautès-Fridman C, Fridman WH. Tertiary lymphoid structures in cancer and beyond. *Trends Immunol*. 2014 Nov;35(11):571-80.

Helmink BA, Reddy SM, Gao J, Zhang S, Basar R, Thakur R, Yizhak K, Sade-Feldman M, Blando J, Han G, Gopalakrishnan V, Xi Y, Zhao H, Amaria RN, Tawbi HA, Cogdill AP, Liu W, LeBleu VS, Kugeratski FG, Patel S, Davies MA, Hwu P, Lee JE, Gershenwald JE, Lucci A, Arora R, Woodman S, Keung EZ, Gaudreau PO, Reuben A, Spencer CN, Burton EM, Haydu LE, Lazar AJ, Zapassodi R, Hudgens CW, Ledesma DA, Ong S, Bailey M, Warren S, Rao D, Krijgsman O, Rozeman EA, Peeper D, Blank CU, Schumacher TN, Butterfield LH, Zelazowska MA, McBride KM, Kalluri R, Allison J, Petitprez F, Fridman WH, Sautès-Fridman C, Hacohen N, Rezvani K, Sharma P, Tetzlaff MT, Wang L, Wargo JA. B cells and tertiary

lymphoid structures promote immunotherapy response. *Nature*. 2020 Jan;577(7791):549-555.

Mahmoud SM, Lee AH, Paish EC, Macmillan RD, Ellis IO, Green AR. The prognostic significance of B lymphocytes in invasive carcinoma of the breast. *Breast Cancer Res Treat*. 2012 Apr;132(2):545-53.

Petitprez F, de Reyniès A, Keung EZ, Chen TW, Sun CM, Calderaro J, Jeng YM, Hsiao LP, Lacroix L, Bougoüin A, Moreira M, Lacroix G, Natario I, Adam J, Lucchesi C, Laizet YH, Toulmonde M, Burgess MA, Bolejack V, Reinke D, Wani KM, Wang WL, Lazar AJ, Roland CL, Wargo JA, Italiano A, Sautès-Fridman C, Tawbi HA, Fridman WH. B cells are associated with survival and immunotherapy response in sarcoma. *Nature*. 2020 Jan;577(7791):556-560.

Savas, P., Virassamy, B., Ye, C., Salim, A., Mintoff, C.P., Caramia, F., Salgado, R., Byrne, D.J., Teo, Z.L., Dushyanthen, S., et al. (2018). Single-cell profiling of breast cancer T cells

Tan C, Mueller JL, Noviski M, Huizar J, Lau D, Dubinin A, Molofsky A, Wilson PC, Zikherman J. Nur77 Links Chronic Antigen Stimulation to B Cell Tolerance by Restricting the Survival of Self-Reactive B Cells in the Periphery. *J Immunol*. 2019 May 15;202(10):2907-2923.

reveals a tissue-resident memory subset associated with improved prognosis. *Nature medicine* 24, 986-993.

Zikherman J, Parameswaran R, Weiss A. Endogenous antigen tunes the responsiveness of naive B cells but not T cells. *Nature*. 2012 Sep 6;489(7414):160-4.

REVIEWER COMMENTS

Reviewer #1 (Remarks to the Author):

manuscript is much improved.

Please make sure the genomic data is really available to other researchers.

Reviewer #2 (Remarks to the Author):

The revised manuscript is significantly improved, and many of my concerns have been addressed. The narrative is generally easier to follow. A major strength of the manuscript continues to be the focus on TIL-B, given these cells are largely understudied. In addition to presenting their scRNAseq data, the authors now provide an excellent summary of the relevant TIL-B literature. However, some concerns remain:

1. As mentioned in my first review, the study still lacks any major discoveries. As the title suggests, it really is an 'atlas' of single cell sequencing data. As such, the impact of this work could be short-lived, given its reliance on a new technical platform.
2. The study could be strengthened by including a reasonably in-depth histological analysis of matched tissues. For example, the authors infer that some samples contained tertiary lymphoid structures, but they did not confirm this histologically. The stromal TIL scores in Fig 1 are inadequate to resolve such questions.
3. Figures 3D and 4D are difficult to understand. What do the X and Y axes represent? The description in the text (lines 199-202) is very confusing.
4. In general, the linkages between the transcriptomic and BCR data are underdeveloped. With deeper analysis, it seems there is an opportunity to learn much more about how TIL-B responses are initiated and mature in tumor tissue.
5. As mentioned in the first reviews, the lack of description/discussion of other immune cell types is a major shortcoming of this study. The authors have data for the entire CD45+ compartment, yet only discuss TIL-B. The study could be greatly improved by describing this additional data, as it would provide physiological context for the TIL-B results. For example, they mention that Tfh cells were found, but what about other CD4 and CD8 phenotypic subsets?
6. The English writing is significantly improved, however the article still requires extensive editing.

Reviewer #3 (Remarks to the Author):

The authors addressed most of my concerns in the revision. However, just using CD3 and CD20 antibodies in IHC staining is not enough to define TLS structures. Please tone down the conclusion on this part. Also, the positions of some of the symbols (the three highest points for TNBC and HER2 groups) in Figure S1B seems higher from what were included in the first version while the baseline seems dropped to close to 0 where it was close to 10,000. The authors should explain these changes.

Reviewer #4 (Remarks to the Author):

Here Hu et al present a single-cell atlas of tumour infiltrating B cells from breast cancer patients. Cells are profiled for transcriptome, BCRs and TCRs (although the latter is not considered in this study). The authors compare the tumour infiltrating B cells with circulating ones and argue that these profiles have prognostic value. Through functional analyses the various mechanisms for how B cells contribute to the immunogenic response are proposed. The authors present a rich dataset on B cells from both PBMCs and tumours and characterize the various sub-populations. Based on the discovery of naive and memory B-cell signatures, they show that it is associated with increased survival rates in cancer patients. They also investigate data from other cancers to present evidence that the same type of TIL-B cells are likely found in other tumour types as well.

I have the following major concerns:

- 1) I am a little bit concerned about the batch correction procedure. First of all, did you look at PCA and tSNE plots prior to batch correction to evaluate to what extent it was needed in the first place? Second, what is the motivation for merging all 20 samples? By merging samples from PBMCs and tumours of different types, then you are essentially eliminating any differences due to real biological differences. I can understand that you would want to merge the TNBC samples from different patients, but why merge them with the LABCs? I think that the authors need to take more care here, motivate why the mergers were done in the way they were, and make sure that important biological differences were not eliminated.
- 2) For Fig 2A (and other similar plots), have you corrected the p-values for multiple hypotheses?
- 3) What test was used for Fig 2B? Please report magnitude of effect as well. Non-germline switched (middle panel) look very similar, yet are reported as significantly different
- 4) On line 155 it is claimed that the mutational spectrum are similar between PBMCs and TILs. However, from fig S4D, it looks like the two densities are significantly different. Could the authors please clarify this discrepancy?
- 5) It is not clear to me why a t-test was used for figure 2D and why the smoothed densities are plotted. Why not plot the CDFs instead and use a K-S test to test for differences?
- 6) The authors need to use an appropriate statistical test to back up the claim that the distribution between the B-cell clusters are different between TNBCs and PBMCs (line 214).

Minor issues:

On line 547 it is reported that the doublet rate is estimated at 14%. How does this compare to the rate estimated from the transcriptome analysis? If it differs by a lot, could the authors comment on why this might be the case?

*line 125 "nature killer" -> natural killer

*line 142 "in rest of analysis" -> in the rest of the analysis

*line 149, 188 agree -> agreement

*line 155 spectrums -> spectra

*line 496, 498, define X%

*line 503 remained -> retained

*line 538 analysis -> analyses

Reviewer #1 (Remarks to the Author):

manuscript is much improved.

Please make sure the genomic data is really available to other researchers.

We have deposited all sequencing data of this manuscript in BIG Data Center, Beijing Institute of Genomics, Chinese Academy of Sciences (<https://bigd.big.ac.cn/gsa>) with accession number CRA002400 for scRNA-seq data and CRA003047 for WES data. Following the instruction of Ministry of Science and Technology (MOST) in China, data of human genetic resources must be put on record before it can be released to the public. We have successfully submitted our application to MOST, and sent all our data by express package to them for review, together with various necessary documents. We are expecting to get their approval soon.

Reviewer #2 (Remarks to the Author):

The revised manuscript is significantly improved, and many of my concerns have been addressed. The narrative is generally easier to follow. A major strength of the manuscript continues to be the focus on TIL-B, given these cells are largely understudied. In addition to presenting their scRNAseq data, the authors now provide an excellent summary of the relevant TIL-B literature. However, some concerns remain:

1. As mentioned in my first review, the study still lacks any major discoveries. As the title suggests, it really is an ‘atlas’ of single cell sequencing data. As such, the impact of this work could be short-lived, given its reliance on a new technical platform.

We partially agree with reviewer’s comment. However, the atlas of TIL-B single cell sequencing data in this study provides a foundation for further studies on B cell tumor immunology, which would be broadly appreciated.

2. The study could be strengthened by including a reasonably in-depth histological analysis of matched tissues. For example, the authors infer that some samples contained tertiary lymphoid structures, but they did not confirm this histologically. The stromal TIL scores in Fig 1 are inadequate to resolve such questions.

Thanks reviewer’s suggestion! To confirm the tertiary lymphoid structures histologically, we conducted immunohistochemistry using CD20, CD3, CD21, and PNA antibodies and multiplex immunofluorescence for CD20/CD3/CD21 and CD20/CD3/Ki67 in the revised manuscript (Fig S12).

3. Figures 3D and 4D are difficult to understand. What do the X and Y axes represent? The description in the text (lines 199–202) is very confusing.

We are very sorry that we did not clearly describe it in previous manuscript. The X and Y axes represent different B cell clusters, and the intersections between X and Y axes represent the shared B cell clones (same germline panel) within the same B cell cluster or between the two B cell clusters, or the shared B cell clones with exactly the same IgH sequence (same IgH panel) within the same B cell cluster or between the two B cell clusters. To describe it more clearly, we updated the text in the revised manuscript with “in TNBC infiltrated B cells, memory B cells had the most clones which share the same inferred IgH germline, and plasma cells had the most clones with the same detected IgH sequence”.

4. In general, the linkages between the transcriptomic and BCR data are underdeveloped. With deeper analysis, it seems there is an opportunity to learn much more about how TIL-B responses are initiated and mature in tumor tissue.

Indeed, this was what we initially expected and also had extensively tried. However, we found that much larger sample size might be necessary to get reliable and conclusive results. This might be relatively easier in mouse models.

5. As mentioned in the first reviews, the lack of description/discussion of other immune cell types is a major shortcoming of this study. The authors have data for the entire CD45+ compartment, yet only discuss TIL-B. The study could be greatly improved by describing this additional data, as it would provide physiological context for the TIL-B results. For example, they mention that Tfh cells were found, but what about other CD4 and CD8 phenotypic subsets?

Thanks reviewer’s suggestion! We analyzed the paired T cell single-cell RNA sequencing and TCR sequencing data and added FigS25-S30 in the revised manuscript. Related results were also added as supplementary text.

6. The english writing is significantly improved, however the article still requires extensive editing.

We carefully polished the whole manuscript again and marked those updated part in red color.

Reviewer #3 (Remarks to the Author):

The authors addressed most of my concerns in the revision. However, just using CD3 and CD20 antibodies in IHC staining is not enough to define TLS structures. Please tone down the conclusion on this part. Also, the positions of some of the symbols (the three highest points for TNBC and HER2 groups) in Figure S1B seems higher from what were included in the first version while the baseline seems dropped to close to 0 where it was close to 10,000. The authors should explain these changes.

To confirm the tertiary lymphoid structures histologically, we conducted immunohistochemistry using CD20, CD3, CD21, and PNA_d antibodies and fluorescent multiplexed immunohistochemistry for CD20/CD3/CD21 and CD20/CD3/Ki67 in the revised manuscript (Fig S12).

Thank you very much for pointing this out. We made a mistake during format conversion of Figure S1B. As showed in resource data excel uploaded along with manuscript, the baseline was close to 0 and the biggest value was 35,382. We updated Figure S1B in the revised manuscript.

Reviewer #4 (Remarks to the Author):

Here Hu et al present a single-cell atlas of tumour infiltrating B cells from breast cancer patients. Cells are profiled for transcriptome, BCRs and TCRs (although the latter is not considered in this study). The authors compare the tumour infiltrating B cells with circulating ones and argue that these profiles have prognostic value. Through functional analyses the various mechanisms for how B cells contribute to the immunogenic response are proposed. The authors present a rich dataset on B cells from both PBMCs and tumours and characterize the various sub-populations. Based on the discovery of naive and memory B-cell signatures, they show that it is associated with increased survival rates in cancer patients. They also investigate data from other cancers to present evidence that the same type of TIL-B cells are likely found in other tumour types as well.

I have the following major concerns:

1) I am a little bit concerned about the batch correction procedure.

First of all, did you look at PCA and tSNE plots prior to batch correction to evaluate to what extent it was needed in the first place?

Second, what is the motivation for merging all 20 samples? By merging samples from PBMCs and tumours of different types, then you are essentially eliminating any differences due to real biological differences. I can understand that you would want to merge the TNBC samples from different patients, but why merge them with the LABCs? I think that the authors need to take more care here, motivate why the mergers were done in the way they were, and make sure that important biological differences were not eliminated.

We checked tSNE plots prior to batch correction. In the tSNE plot, cells mostly clustered together according to the scRNA-sequencing library batches (left panel), even for the same cell types (right panel), indicating that batch effect was detected and batch correction was needed.

To compare the composition of canonical immune cell types among all types of breast cancers, we merged all 20 samples in Fig 1B with very low clustering resolution. We agree with that differences could be eliminated by batch correction, especially when define of small cell

populations is necessary. Instead of merging different tumor types together, we only merged TNBC samples for the following B cell analyses to prevent eliminating important biological differences.

2) For Fig 2A (and other similar plots), have you corrected the p-values for multiple hypotheses?

We did not correct p-values for multiple hypotheses in Fig2A, Fig3C and FigS11, since this has not been done in other publications (such as Zheng et al. Cell 2017, Sathe A et al. Clin Cancer Res 2020 and Oh DY et al. Cell 2020) too.

Indeed, as shown in the following table, some of the p-values were not statistically significant after such corrections. Therefore, in the revised manuscript, we described the differences more carefully and did not emphasize “significantly”.

		P	P-adjust(FDR)
Fig 2A	IgM	0.04	0.12
	IgD	0.12	0.16
	IgG	0.06	0.12
	IgA	0.5	0.5
Fig3C	Naïve	0.02	0.04
	Bmem	0.03	0.04
	Plasma	0.25	0.25
	CD14	0.01	0.04
FigS11	C1	0.0048	0.04
	C2	0.13	0.21
	C3	0.98	0.98
	C4	0.47	0.56
	C5	0.07	0.15
	C6	0.01	0.04
	C7	0.03	0.10
	C8	0.62	0.67
	C9	0.1	0.19
	C10	0.22	0.30
	C11	0.23	0.30
	C12	0.04	0.10
	C13	0.0084	0.04

3) What test was used for Fig 2B? Please report magnitude of effect as well. Non-germline switched (middle panel) look very similar, yet are reported as significantly different

We used student t-test for Fig 2B. The cell numbers and average mutation frequencies \pm standard deviation were shown in the following table. Although the Non-germline switched (middle panel) look very similar (6.39% \pm 3.45% VS 6.73% \pm 3.45%, with magnitude of TNBC/PBMC=0.95), the T-test p value is 0.0023 and wilcoxon test p value is 0.00367 for it. We added the following table to the source data of Fig 2B.

	All B cells		Non-germline switched		Non-germline non-switched	
Tissue	TNBC	PBMC	TNBC	PBMC	TNBC	PBMC
Cell number	3695	8037	1689	2246	1265	2238
Magnitude	TNBC/PBMC=1.76		TNBC/PBMC=0.95		TNBC/PBMC=1.27	
Mut_freq	5.43%±3.59%	3.08%±3.79%	6.39%±3.45%	6.73%±3.45%	4.84%±2.96%	3.81%±3.04%

4) On line 155 it is claimed that the mutational spectrum are similar between PBMCs and TILs. However, from fig S4D, it looks like the two densities are significantly different. Could the authors please clarify this discrepancy?

In Fig S4D, the heatmap showed similar mutational spectrum between B cells from PBMCs and TNBCs. The p-values were listed below, and there were no significant difference between TNBC and PBMC for all of the mutation types except G->T (p=0.02, two-tailed paired Student's t-test). We changed the sentence on line155 to "overall, there was no significant difference between TNBC and PBMC for all of the mutation types except G->T" (line 161 in the revised manuscript)

5) It is not clear to me why a t-test was used for figure 2D and why the smoothed densities are plotted. Why not plot the CDFs instead and use a K-S test to test for differences?

Thanks reviewer's suggestion! We used K-S test (p= 3.98 x 10⁻⁴) to test the differences instead of t-test in the revised manuscript. We also tried CDFs as below. We followed the similar format (Meng et al. 2017, NBT) to show that B cell clones in TNBCs had more mutations than clones in PBMCs in Fig 2D.

6) The authors need to use an appropriate statistical test to back up the claim that the distribution between the B-cell clusters are different between TNBCs and PBMCs (line 214).

We claimed that the average percentages of different B cell subgroups were different between TNBCs and PBMCs, and we used t-test to back up this (FigS11). We also noticed that some B cell groups in TNBCs (C7, C8, C9, C10, and C11) showed larger variation than PBMCs (Fig 4C). As this observation is preliminary, we revised the main text as “The composition of those B cell clusters was different between tumor and PBMC samples”

Minor issues:

On line 547 it is reported that the doublet rate is estimated at 14%. How does this compare to the rate estimated from the transcriptome analysis? If it differs by a lot, could the authors comment on why this might be the case?

The estimated doublet rate for the transcriptome data was around 7% in our study, calculated based on the reported doublet rates for libraries with various cell numbers [Grace X.Y. Zheng et al., 2017, Nat Commun].

The doublet rate was estimated at 14% for our BCR data. However, this higher doublet rate might be caused by our stringent cutoff to remove all possible doublets. To remove potential doublet as clean as we can, we discarded nearly all B cells detected with multi-productive IgH alleles. In fact, some of cells with multi-productive IgH alleles might be caused by full length IgH allele assemble algorithm using short sequencing reads.

*line 125 “nature killer” -> natural killer

*line 142 “in rest of analysis” -> in the rest of the analysis

*line 149, 188 agree -> agreement

*line 155 spectrums -> spectra

*line 496, 498, define X%

*line 503 remained -> retained

*line 538 analysis -> analyses

Thanks for the reviewer to point them out. We corrected them in the revised manuscript. We carefully revised the whole manuscript again and marked those updated parts in red color.

In revised manuscript, we define the X in a more straightforward way, “where X was inferred as $X=(0.000879*N+0.702)*0.01*N$ with linear fitting using the 10x platform data from supplementary Fig 1a³⁶, where N represents the cell number detected by Cell Ranger”

Reference

Zheng C, Zheng L, Yoo JK, Guo H, Zhang Y, Guo X, Kang B, Hu R, Huang JY, Zhang Q, Liu Z, Dong M, Hu X, Ouyang W, Peng J, Zhang Z. Landscape of Infiltrating T Cells in Liver Cancer Revealed by Single-

Cell Sequencing. *Cell*. 2017 Jun 15;169(7):1342-1356.e16.

Sathe A, Grimes SM, Lau BT, Chen J, Suarez C, Huang RJ, Poultsides G, Ji HP. Single-Cell Genomic Characterization Reveals the Cellular Reprogramming of the Gastric Tumor Microenvironment. *Clin Cancer Res*. 2020 Jun 1;26(11):2640-2653.

Oh DY, Kwek SS, Raju SS, Li T, McCarthy E, Chow E, Aran D, Ilano A, Pai CS, Rancan C, Allaire K, Burra A, Sun Y, Spitzer MH, Mangul S, Porten S, Meng MV, Friedlander TW, Ye CJ, Fong L. Intratumoral CD4⁺ T Cells Mediate Anti-tumor Cytotoxicity in Human Bladder Cancer. *Cell*. 2020 Jun 25;181(7):1612-1625.e13.

Meng W, Zhang B, Schwartz GW, Rosenfeld AM, Ren D, Thome JJC, Carpenter DJ, Matsuoka N, Lerner H, Friedman AL, Granot T, Farber DL, Shlomchik MJ, Hershberg U, Luning Prak ET. An atlas of B-cell clonal distribution in the human body. *Nat Biotechnol*. 2017 Sep;35(9):879-884.

Zheng GX, Terry JM, Belgrader P, Ryvkin P, Bent ZW, Wilson R, Ziraldo SB, Wheeler TD, McDermott GP, Zhu J, Gregory MT, Shuga J, Montesclaros L, Underwood JG, Masquelier DA, Nishimura SY, Schnall-Levin M, Wyatt PW, Hindson CM, Bharadwaj R, Wong A, Ness KD, Beppu LW, Deeg HJ, McFarland C, Loeb KR, Valente WJ, Ericson NG, Stevens EA, Radich JP, Mikkelsen TS, Hindson BJ, Bielas JH. Massively parallel digital transcriptional profiling of single cells. *Nat Commun*. 2017 Jan 16;8:14049.

REVIEWERS' COMMENTS

Reviewer #2 (Remarks to the Author):

While this manuscript is significantly improved over the prior two versions, it still lacks any major discoveries that significantly advance our understanding of TIL-B in human cancer, or our understanding of the general immunobiology of breast cancer. It is well established that breast tumors harbour TIL-B and that TNBC are "hotter" than other breast ca subtypes. Moreover, most studies to date agree that TIL-B are prognostically favourable in breast cancer. While this atlas of molecular profiles of TIL-B will undoubtedly be of use to the community at large (along with similar data from other single cell seq studies), the field has moved beyond the stage where simply reporting single-cell sequencing data alone is sufficient justification for publication in high impact journals.

Reviewer #3 (Remarks to the Author):

Authors have addressed my concerns.

Reviewer #4 (Remarks to the Author):

The authors have addressed all of my comments and in addition they have substantially improved the writing, making the manuscript a much more pleasant read.

Reviewer #2 (Remarks to the Author):

While this manuscript is significantly improved over the prior two versions, it still lacks any major discoveries that significantly advance our understanding of TIL-B in human cancer, or our understanding of the general immunobiology of breast cancer. It is well established that breast tumors harbour TIL-B and that TNBC are "hotter" than other breast ca subtypes. Moreover, most studies to date agree that TIL-B are prognostically favourable in breast cancer. While this atlas of molecular profiles of TIL-B will undoubtedly be of use to the community at large (along with similar data from other single cell seq studies), the field has moved beyond the stage where simply reporting single-cell sequencing data alone is sufficient justification for publication in high impact journals.

We partially agree with reviewer's comment. However, the atlas of TIL-B single cell sequencing data in this study provides a foundation for further studies on B cell tumor immunology, which would be broadly appreciated.